# EDITREWARD: A HUMAN-ALIGNED REWARD MODEL FOR INSTRUCTION-GUIDED IMAGE EDITING

[†]**Keming Wu**[1,2*]**, Sicong Jiang**[3,4*]**, Max Ku**[1]**, Ping Nie**[5]**, Minghao Liu**[3]**, **[†]**Wenhu Chen**[1]

[1]University of Waterloo, [2]Tsinghua Univerisity, [3]2077AI, [4]McGill University, [5]Independent
{wukeming0608@gmail.com, wenhuchen@uwaterloo.ca}

✦ https://tiger-ai-lab.github.io/EditReward

## ABSTRACT

Recently, we have witnessed great progress in image editing with natural language instructions. Several closed-source models like GPT-Image-1, Seedream, and Google-Nano-Banana have shown highly promising progress. However, the open-source models are still lagging. The main bottleneck is the lack of a reliable reward model to scale up high-quality synthetic training data. To address this critical bottleneck, we built EDITREWARD, trained with our new large-scale human preference dataset, meticulously annotated by trained experts following a rigorous protocol containing over 200K preference pairs. EDITREWARD demonstrates superior alignment with human preferences in instruction-guided image editing tasks. Experiments show that EDITREWARD achieves state-of-the-art human correlation on established benchmarks such as GenAI-Bench, AURORA-Bench, ImagenHub, and our new EDITREWARD-BENCH, outperforming a wide range of VLM-as-judge models. Furthermore, we use EDITREWARD to select a high-quality subset from the existing noisy ShareGPT-4o-Image dataset. We train Step1X-Edit on the selected subset, which shows significant improvement over training on the full set. This demonstrates EDITREWARD's ability to serve as a reward model to scale up high-quality training data for image editing. EDITRE-WARD with its training dataset will be released to help the community build more high-quality image editing training datasets to catch up with the frontier ones.

## 1 INTRODUCTION

Instruction-guided image editing is an important task to enable intuitive and fine-grained image modifications through natural language instructions (Brooks et al., 2023; Zhang et al., 2024a; Zhao et al., 2024; Wei et al., 2024). Closed-source models like GPT-Image-1 (OpenAI, 2025), Seedream (Gao et al., 2025), and Google's Nano Banana (Google DeepMind, 2025) have made marvelous strides on this task. The progress is driven partially by their high-quality in-house private training dataset. Existing open-source image editing datasets like ImgEdit (Ye et al., 2025), HQ-Edit (Hui et al., 2024), GPT-Image-Edit-1.5M (Wang et al., 2025d), UltraEdit (Zhao et al., 2024), and OmniEdit (Wei et al., 2024) are all produced with automatic data synthesis pipelines and filtered with different rewards.

Commonly used rewards are mainly divided into three categories: (1) Perceptual scores like LPIPS (Zhang et al., 2018) fail to capture semantic alignment with user instructions, (2) Feature scores like CLIP (Hessel et al., 2021) fail to capture editing semantics, (3) VLM-as-a-judge like VIEScore (Ku et al., 2023; Jiang et al., 2024; Wang et al., 2025b) uses general-purpose Vision-Language Models (VLM), which are not optimized for rewarding image editing tasks. Therefore, these ad-hoc rewards show weak alignment with human preference in the image editing task. To build more aligned rewards, a line of work proposes to fine-tune general-purpose VLMs (for instance Qwen2.5-VL) to reward models. However, some of them rely on noisy, crowd-sourced preference annotations (Lin et al., 2024; Xu et al., 2024), which are often plagued by inconsistency, low inter-annotator agreement. The others adopt pseudo-labels generated by proprietary, closed-source models (Wei et al., 2024; Wu et al., 2025c), creating highly noisy and biased labels. These trained

---

*These authors contributed equally.

reward models still fall short in providing enough reward signals to scale up high-quality image editing datasets. A high quality image editing dataset is desired to build a good reward model.

In this paper, we introduce EDITREWARD, a human-aligned reward model powered by a high-quality dataset for instruction-guided image editing. We first construct EDITREWARD-DATA, a large-scale, high-fidelity preference dataset for instruction-guided image editing. It comprises over 200K manually annotated preference pairs, covering a diverse range of edits produced by seven state-of-the-art models across twelve distinct sources. Every preference annotation in EDITREWARD-DATA was curated by trained annotators following a rigorous and standardized protocol, ensuring high alignment with considered human judgment and minimizing label noise. Using this dataset, we train the reward model EDITREWARD to score instruction-guided image edits. To rigorously assess EDITREWARD and future models, we also introduce EDITREWARD-BENCH, a new benchmark built upon our high-quality annotations, which includes more difficult multi-way preference prediction.

Experimental results show that EDITREWARD achieves state-of-the-art performance on several benchmarks. On GenAI-Bench (Jiang et al., 2024), our model obtains a score of 65.72, significantly outperforming other leading VLM judges such as GPT-5 (59.61). Similarly, on AURORA-Bench (Krojer et al., 2024), EDITREWARD scores 63.62, showing a substantial gain over OpenAI-GPT-4o (50.81). While demonstrating competitive performance on ImagenHub (Ku et al., 2024) with a score of 35.20, it is on our proposed EDITREWARD-BENCH where the fine-grained capabilities of top models are most clearly discerned. This not only validates the superiority of our model but also demonstrates that EDITREWARD-BENCH provides a more reliable and challenging evaluation. We further study the potential of EDITREWARD to select the high-quality subset from noisy candidates, which can be used to train next-generation image editing models. Specifically, we adopt EDITREWARD to select the top 20K subset from ShareGPT-4o-Image (Chen et al., 2025a) and use the subset to fine-tune Step1X-Edit (Liu et al., 2025b). We observe significant improvement by training on the subset over training on the full set. On GEdit-Bench, the overall score increases from 6.7/10 (full-set) to 7.1/10 (subset), making it on par with Doubao-Edit (Wang et al., 2025c). This experiment demonstrates its high potential to work as a reward model for future research.

In summary, our primary contributions are: **(1)** We construct and release EDITREWARD-DATA, a large-scale (200K) preference dataset for image editing, distinguished by its high-quality manual annotations and diversity of sources. **(2)** We train and release EDITREWARD, a VLM-based reward model trained on EDITREWARD-DATA that demonstrates superior alignment with human preferences. **(3)** We propose EDITREWARD-BENCH, a new benchmark featuring a more challenging multi-way preference ranking task that provides a more robust evaluation of reward models.

## 2 EDITREWARD-DATA

### 2.1 THE EDITREWARD-DATA CONSTRUCTION

EDITREWARD-DATA contains 9557 instruction–image pairs collected from six established editing benchmarks: GEdit-Bench (606) (Liu et al., 2025b), ImgEdit-Bench (737) (Ye et al., 2025), MagicBrush (1,053) (Zhang et al., 2024a), AnyEdit (1,250) (Yu et al., 2025), EmuEdit (5,611) (Sheynin et al., 2024), and an internal set (300). This aggregation ensures broad coverage of semantically grounded and executable editing instructions. For each instruction, we generated 12 candidate images using six state-of-the-art models: Step1X-Edit (Liu et al., 2025b), Flux-Kontext (BlackForestLabs et al., 2025), Qwen-Image-Edit (Wu et al., 2025a), BAGEL (Deng et al., 2025), Ovis-U1 (Wang et al., 2025a), and OmniGen2 (Wu et al., 2025b), with multiple random seeds to avoid model bias. Seven candidates were randomly sampled for human evaluation. Annotators scored each image on a 4-point Likert scale (1 = Poor and 4 = Excellent) along two dimensions: **Instruction Following** (semantic accuracy, completeness, and no unprompted changes) and **Visual Quality** (plausibility, artifact-free rendering, and aesthetics). This rubric yields more informative labels than single-score schemes. Details of the annotation protocol and quality-control process are provided in the Appendix A.2. Comprehensive statistics of the dataset are provided in Table 1 and Figure 2. EDITREWARD-DATA is unique in combining large-scale, expert human annotation and a multi-dimensional scoring rubric, making it a strong foundation for training editing reward models. More representative examples from EDITREWARD-DATA are shown in the Appendix A.10. The IAA results in Table 9 provide a critical quantitative assessment of our expert annotation quality (Fleiss, 1971). We highlight

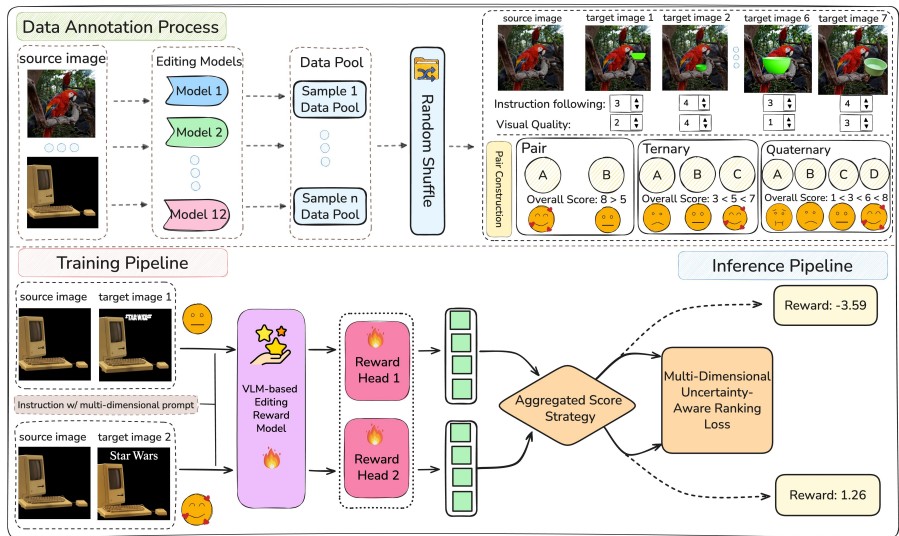

Figure 1: An overview of our framework, illustrating the construction of the EDITREWARD-DATA and the subsequent training of our reward model, EDITREWARD. **Top:** The data pipeline, where we generate a diverse candidate pool from multiple state-of-the-art models and collect multi-dimensional human preference annotations. **Bottom:** The model pipeline, where EDITREWARD is optimized on EDITREWARD-DATA using our proposed Multi-Dimensional Uncertainty-Aware Ranking Loss for training, followed by its use in inference.

the values derived from Krippendorff's Alpha ($\alpha$) (Krippendorff, 2011), which is the most appropriate metric as it correctly models the ordinal nature of our 4-point Likert scale. The $\alpha$ scores of **0.668** for Instruction Following (IF) and **0.597** for Visual Quality (VQ) establish a strong, quantified baseline for human consistency. Crucially, the observed difference ($IAA_{IF} > IAA_{VQ}$) provides empirical validation for our core contribution: it confirms that the VQ dimension is inherently more subjective than IF. This validates our design choice to use a multi-dimensional rubric and a multi-head reward model, as a single holistic score would obscure this critical difference in human variance.

Table 1: The comparison of different generative preference datasets and benchmarks.

| Dataset | Scale | Task Focus | Annotation | Eval. Dims. | Limitation / Caveat |
|---|---|---|---|---|---|
| ImageRewardDB(Xu et al., 2024) | ~137K | Visual Generation | Human | Single | Noise, limited diversity |
| VisionPrefer(Wu et al., 2025c) | ~1.2M | Visual Generation | Model | Multiple | Model bias, synthetic prefs |
| GenAI-Bench(Jiang et al., 2024) | ~1.6K | Generation / Editing | Human | Single | Small scale |
| HIVE(Zhang et al., 2024b) | ~3.6K | Instructional Editing | Human | Single | Small reward set |
| ADIEE(Chen et al., 2025b) | ~100K | Instructional Editing | Model | Single | Synthetic labels, model bias |
| HPSv3(Ma et al., 2025) | ~1.17M | Visual Generation | Human | Single | Generalization limits |
| **EDITREWARD-DATA** | ~200K | Instructional Editing | Human | Multiple | Fine-grained supervision |

| Benchmark | Scale | Annotation | Eval. Dims. | Multi-Way Preference | Pair-Wise | Point-Wise |
|---|---|---|---|---|---|---|
| GenAI-Bench-Edit(Jiang et al., 2024) | ~900 | Human | Multiple | 2-way | ✓ | – |
| AURORA-Bench-Edit(Krojer et al., 2024) | ~1.6K | Human | Single | 2-way | ✓ | ✓ |
| ImagenHub-Edit(Ku et al., 2024) | ~1.4K | Human | Multiple | – | – | ✓ |
| **EDITREWARD-BENCH** | ~1.5K | Human Cross-check | Multiple | 2/3/4-way | ✓ | ✓ |

Table 2: Inter-Annotator Agreement (IAA) Metrics for Expert Annotations. The robust Krippendorff Alpha ($\alpha$) values confirm the high reliability of our multi-dimensional expert scoring.

| Dataset | Fleiss' Kappa (IF) | Fleiss' Kappa (VQ) | Krippendorff Alpha (IF) | Krippendorff Alpha (VQ) |
|---|---|---|---|---|
| EDITREWARD-DATA | 0.4157 | 0.3203 | 0.6762 | 0.5720 |
| EDITREWARD-BENCH | 0.3962 | 0.3157 | 0.6623 | 0.6114 |
| All data | 0.3994 | 0.3111 | 0.6685 | 0.5972 |

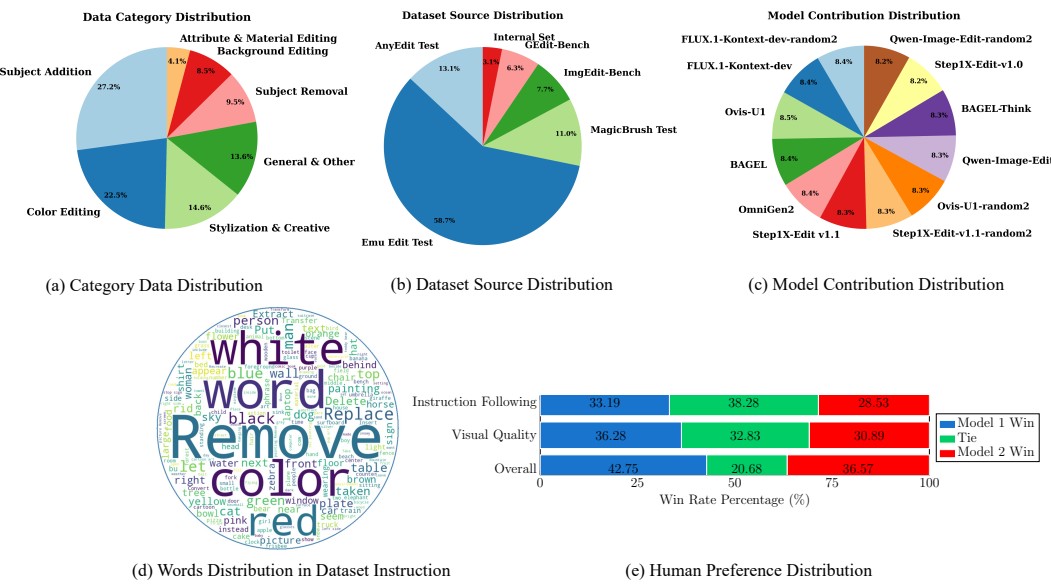

Figure 2: Statistics of our EDITREWARD-DATA and EDITREWARD-BENCH.

## 2.2 THE EDITREWARD-BENCH CONSTRUCTION

EDITREWARD-BENCH is designed to provide a more robust evaluation of image editing reward models than existing suites. We curated 500 high-quality groups from the EDITREWARD-DATA candidate pool, covering diverse editing categories. Each group was annotated by three independent experts using the same two-dimensional rubric (instruction following and visual quality) described in Section 2.1. We prioritized challenging cases where competing edits had small score differences to increase the discriminative power of the benchmark. The key innovation of EDITREWARD-BENCH is a **multi-way preference comparison** protocol that extends beyond pairwise judgments. Evaluation units include ternary (A, B, C) and quaternary (A, B, C, D) tuples, with correctness defined by simultaneously predicting all pairwise relations within the tuple. This strict criterion provides a more comprehensive and reliable test of ranking consistency than traditional pairwise accuracy. We benchmark a wide range of models on EDITREWARD-BENCH, and results are reported in Section 4. More Details of the construction of EDITREWARD-BENCH are provided in the Appendix A.3.

# 3 EDITREWARD

## 3.1 ARCHITECTURE

Inspired by the success of VLMs as powerful feature extractors, we leverage a VLM as the backbone for our reward model. The task of image editing evaluation is inherently tri-modal, requiring joint reasoning over a source image ($I_s$), a textual prompt ($P$), and an edited image ($I_e$). Our model is trained on human preference data, which consists of pairs of edited images, ($I_{e,1}, I_{e,2}$), generated from the same ($I_s, P$) context.

Our reward model consists of two components: a multimodal backbone, $\mathcal{H}_\psi$ (either Qwen2.5-VL (Bai et al., 2023) or Mimo-VL (Yue et al., 2025)), which computes a latent representation of the edit's quality; and an MLP reward head, $\mathcal{R}_\omega$, which projects this representation to a scalar score. The score $s_i$ for an edited image $I_{e,i}$ is thus given by:

$$s_i = \mathcal{R}_\omega\big(\mathcal{H}_\psi(I_s, P, I_{e,i})\big). \tag{1}$$

Here $\mathcal{H}_\psi$ represents the VLM backbone with parameters $\psi$, and $\mathcal{R}_\omega$ is the MLP reward head with parameters $\omega$. For a preference pair, the scores $s_1$ and $s_2$ are computed using Eq. 1 and are subsequently used in a preference loss function to jointly optimize the parameters $\psi$ and $\omega$.

## 3.2 MULTI-DIMENSIONAL UNCERTAINTY-AWARE RANKING

Prior reward models for generative tasks often fail to account for inconsistencies in human annotations, treating each preference label with equal certainty. This can introduce bias, particularly when judging ambiguous or challenging cases. The HPSv3 framework (Sun et al., 2025) made significant progress in text-to-image evaluation by addressing this issue. Instead of predicting a deterministic score $s$, HPSv3 models the score as a Gaussian distribution $s \sim \mathcal{N}(\mu, \sigma^2)$, thereby capturing the uncertainty inherent in the data. The preference probability $P(I_{e,1} \succ I_{e,2})$ is then computed by integrating over the two reward distributions, leading to a more robust, probabilistic ranking.

Inspired by this, we adapt and extend this uncertainty-aware paradigm for the more complex domain of instruction-guided image editing. Image editing quality is multi-faceted; an edit can be faithful to the instruction but visually unrealistic, or vice versa. To capture this complexity, our EDITREWARD-DATA provides disentangled scores across two distinct dimensions: (1) *Instruction Following* and (2) *Visual Quality*. A single, holistic uncertainty distribution as in HPSv3 is insufficient to model this rich, multi-dimensional feedback.

To this end, we adapt the reward head, $\mathcal{R}_\omega$, using a Multi-Task Learning (MTL) (Crawshaw, 2020) approach. For a single edited image sample $(I_s, P, I_e)$, the reward head no longer outputs a single distribution, but rather a separate Gaussian distribution for each evaluation dimension. Let $d \in \{1, 2\}$ represent the two dimensions. The output for a single sample $i$ is a pair of distributions as formulated in Eq. 2:

$$s_{i,d} \sim \mathcal{N}(\mu_{i,d}, \sigma_{i,d}^2), \quad \text{for } d = 1, 2. \tag{2}$$

This is achieved by having the final layers of the MLP in $\mathcal{R}_\omega$ predict a set of parameters $(\mu_{i,1}, \sigma_{i,1}, \mu_{i,2}, \sigma_{i,2})$ for each input. We explore both separate and shared-parameter heads for the task. To train our model with this multi-dimensional output, we explore two distinct loss:

**Multi-Dimensional Uncertainty-Aware Ranking Loss.** This approach extends the probabilistic ranking framework of HPSv3 (Sun et al., 2025) to our multi-dimensional task. To do so, we must first aggregate the two predicted dimensional mean scores $(\mu_{i,1}, \mu_{i,2})$ for each candidate image $i$ into a single, effective mean score, $\mu_i^{\text{agg}}$. We propose and investigate three distinct aggregation strategies, which can be compactly formulated as Eq. 3:

$$\mu_i^{\text{agg}} = \begin{cases} \min(\mu_{i,1}, \mu_{i,2}) & \text{(Pessimistic Minimum)} \\ \frac{1}{2}(\mu_{i,1} + \mu_{i,2}) & \text{(Balanced Average)} \\ \mu_{i,1} + \mu_{i,2} & \text{(Direct Summation)} \end{cases} \tag{3}$$

The resulting aggregated means for a pair of images, along with their predicted uncertainties ($\sigma^2$), are then used to compute the final preference probability $P(I_h \succ I_l)$ following the probabilistic method from HPSv3. The model is trained by minimizing the negative log-likelihood of the ground-truth preference as in Eq. 4:

$$\mathcal{L}_{\text{rank}} = -\log(P(I_h \succ I_l)). \tag{4}$$

**Aggregated Score Regression.** Alternatively, we frame the training as a direct regression task. This approach leverages the pointwise scores available in our EDITREWARD-DATA dataset by first aggregating the predicted distributions. Given that the sum of two independent Gaussians is also a Gaussian, the aggregated score distribution for a sample $i$ is $s_{i,\text{agg}} \sim \mathcal{N}(\mu_{i,1} + \mu_{i,2}, \sigma_{i,1}^2 + \sigma_{i,2}^2)$. The model is then optimized by minimizing the Mean Squared Error (MSE) between the mean of this aggregated distribution and a transformed sum of the ground-truth scores, $\tilde{z}_{\text{agg}} = \mathcal{T}(z_1 + z_2)$:

$$\mathcal{L}_{\text{reg}} = \mathbb{E}_{(I_s, P, I_e, z_1, z_2) \sim \mathcal{D}} \left[ \|(\mu_{i,1} + \mu_{i,2}) - \tilde{z}_{\text{agg}}\|^2 \right]. \tag{5}$$

This multi-dimensional uncertainty-aware approach allows our model to learn a more nuanced and disentangled representation of edit quality, leveraging the rich supervisory signal in our dataset. Ablation study comparing the loss functions and aggregation strategies is presented in Section 4.6.

## 3.3 DISENTANGLING TIES VIA DIMENSIONAL PREFERENCE.

While standard models like Bradley-Terry model with ties (BTT) treat tied pairs as a single outcome (Liu et al., 2025a), we propose a novel data augmentation strategy to extract a richer supervisory signal from these ambiguous cases. Our key insight is that a tie in overall quality often masks

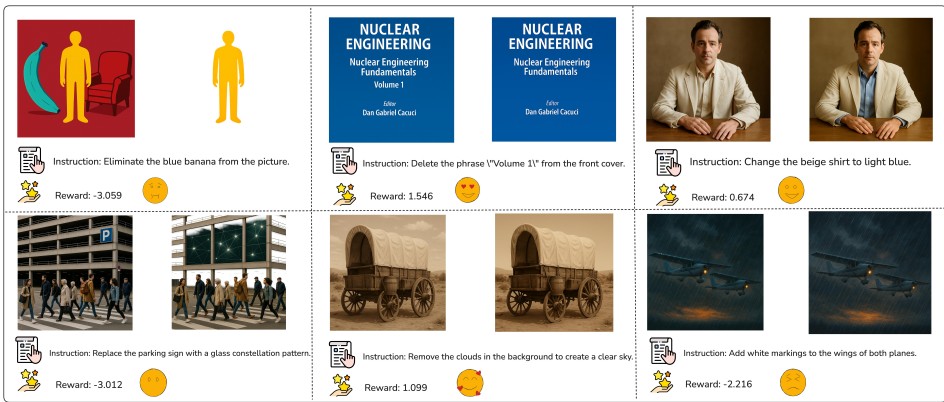

Figure 3: Representative examples of our reward model aligning with human judgments.

complementary dimensional strengths. For instance, one image may excel in *Instruction Following* while the other has superior *Visual Quality*. We leverage this by decomposing each qualifying tie pair $(I_A, I_B)_{\text{tie}}$ into two new training samples with opposing preference labels, $(I_A \succ I_B)$ and $(I_B \succ I_A)$, based on their respective dimensional advantages (Eq. 6). Let $z_{i,d}$ be the ground-truth score for image $i$ on dimension $d$. A tie pair where one image is preferred on the first dimension and the other is preferred on the second (e.g., $z_{A,1} > z_{B,1}$ and $z_{B,2} > z_{A,2}$) is duplicated and relabeled as follows:

$$\text{The pair } (I_A, I_B)_{\text{tie}} \implies \begin{cases} \text{Sample 1 with label: } I_A \succ I_B \\ \text{Sample 2 with label: } I_B \succ I_A \end{cases} \tag{6}$$

This strategy forces the model to reconcile seemingly contradictory signals for the same input pair, pushing it to develop a more granular understanding of nuanced trade-offs. This not only doubles the utility of our annotated tie data but also leads to a more stable training dynamic. As illustrated in Appendix A.7, our tie-disentanglement method results in a smoother training loss curve and more consistent performance gains on the validation set. Figure 3 shows some examples of our reward model giving rewards that are aligned with humans. More failure mode anaylysis of EDITREWARD is shown in th Appendix A.13.

## 4 EXPERIMENTS

### 4.1 IMPLEMENTATION DETAILS

We train our reward model, EDITREWARD, using 200K high-quality pairwise preference samples from our dataset. For our main experimental results, we report performance using two powerful vision-language models as backbones: Qwen2.5-VL-7B and MiMo-VL-7B. To ensure a controlled comparison, all ablation studies are conducted consistently using the Qwen2.5-VL-7B backbone. During training, all parameters of the backbone are unfrozen and set as trainable. The training is performed for 2 epochs on a cluster of 8 NVIDIA A800 GPUs. We follow the hyperparameter configuration to HPSv3, using a learning rate of $2 \times 10^{-6}$ with a cosine learning rate schedule and a warm-up ratio of 0.05. With a per-GPU batch size of 2, the total effective batch size is 16. For preprocessing, all training images are resized to $448 \times 448$ pixels while preserving their original aspect ratios. Additional training details are provided in Appendix A.4.

### 4.2 BENCHMARKS AND BASELINES

We evaluate our approach on a suite of three established public benchmarks and our newly proposed benchmark, designed to provide a more comprehensive assessment of image editing quality.

**Existing Benchmarks.** We utilize ImagenHub (Ku et al., 2024), GenAI-Bench (Jiang et al., 2024), and AURORA-Bench (Krojer et al., 2024). We explicitly confirm that we verified no overlap exists between our EDITREWARD-DATA training set and these evaluation benchmarks. They serve as fully

Table 3: Comprehensive results on public benchmarks and our proposed EDITREWARD-BENCH. Under the EDITREWARD-BENCH results, K denotes the number of candidates in the multi-way preference ranking task. **Bold** marks the best performance, and underline marks the second best.

| Method | GenAI-Bench | AURORA-Bench | Imagen Hub | EDITREWARD-BENCH | | | |
|---|---|---|---|---|---|---|---|
| | | | | K=2 | K=3 | K=4 | Overall |
| Random | 25.90 | 33.43 | – | 25.81 | 11.33 | 1.35 | 13.84 |
| Human-to-Human | – | – | 41.84 | – | – | – | – |
| *Proprietary Models* | | | | | | | |
| GPT-4o | 53.54 | 50.81 | 38.21 | 45.69 | 27.33 | 7.31 | 28.31 |
| GPT-5 | 59.61 | 47.27 | 40.85 | 57.53 | 38.51 | 12.84 | 37.81 |
| Gemini-2.0-Flash | 53.32 | 44.31 | 23.69 | 52.43 | 33.33 | **13.51** | 33.47 |
| Gemini-2.5-Flash | 57.01 | 47.63 | **41.62** | **58.61** | 39.86 | 12.16 | 38.02 |
| *Open-Source VLMs* | | | | | | | |
| Qwen2.5-VL-3B-Inst | 42.76 | 30.69 | -2.54 | 51.07 | 20.27 | 2.71 | 26.86 |
| Qwen2.5-VL-7B-Inst | 40.48 | 38.62 | 18.59 | 52.69 | 24.67 | 3.38 | 29.75 |
| Qwen2.5-VL-32B-Inst | 39.28 | 37.06 | 26.87 | 50.54 | 25.27 | 4.05 | 28.72 |
| MiMo-VL-7B-SFT-2508 | 57.89 | 30.43 | 22.14 | 49.46 | 30.41 | 9.46 | 31.19 |
| ADIEE | 59.96 | 55.56 | 34.50 | – | – | – | – |
| *Reward Models (Ours)* | | | | | | | |
| EDITREWARD (on Qwen2.5-VL-7B) | 63.97 | 59.50 | 36.18 | 56.99 | 36.00 | 10.81 | 36.78 |
| EDITREWARD (on MiMo-VL-7B-SFT) | **65.72** | **63.62** | 35.20 | 56.45 | **42.67** | 11.49 | **38.42** |

independent, held-out testbeds, ensuring a fair and unbiased evaluation of EDITREWARD's generalization. For benchmarks with point-wise annotations like ImagenHub, we measure the Spearman rank correlation to assess alignment with human scores. For ImagenHub, which includes three ratings per sample, we also compute the Human-to-Human correlation as a practical upper bound (Ku et al., 2023). For benchmarks with paired comparisons like GenAI-Bench and the pair-wise split of AURORA-Bench, we report the prediction accuracy. Additional details of the evaluation across different methods are provided in Appendix A.5.

**EDITREWARD-BENCH.** Derived from the held-out test split of our EDITREWARD-DATA dataset, this benchmark provides pair-wise preference labels. We report performance on EDITREWARD-BENCH using overall preference accuracy (pair-wise). We evaluated a wide range of leading models on EDITREWARD-BENCH to establish its utility. This included proprietary models such as GPT-4o, GPT-5, Gemini-2.0-Flash (Hassabis et al., 2024), and Gemini-2.5-Flash (Comanici et al., 2025), as well as prominent open-source VLMs like the Qwen2.5-VL series and MiMo-VL-7B. The experimental results, detailed in Section 4, demonstrate that EDITREWARD-BENCH effectively differentiates between models of varying capabilities and reveals challenges, such as reasoning over multiple candidates, that are not apparent in simpler pairwise benchmarks.

## 4.3 EXPERIMENTAL RESULTS: ALIGNMENT WITH HUMANS

The main results presented in Table 3 establish EDITREWARD as a new state-of-the-art reward model for instruction-guided image editing. Our best model, **EDITREWARD (on MiMo-VL-7B)**, achieves top scores on the primary public benchmarks, obtaining an accuracy of **65.72%** on GenAI-Bench and **63.62%** on AURORA-Bench. This performance surpasses strong proprietary models like GPT-5 (59.61) and the leading open-source method ADIEE (59.96). On the point-wise ImagenHub benchmark, our model remains highly competitive with the best systems available, the Qwen2.5-VL-7B variant achieves a second-best Spearman correlation of 36.18, closely following GPT-4o.

Crucially, our results highlight the profound impact of our training paradigm itself. By applying our methodology to the base Qwen2.5-VL-7B model, we observe a massive performance uplift of over **23 points** on GenAI-Bench (from 40.48% to 63.97%), demonstrating that our framework dramatically enhances a VLM's alignment with human judgments. This capability is further validated on our challenging EDITREWARD-BENCH, where **EDITREWARD (on MiMo-VL-7B)** again achieves the highest score of **38.42%**, outperforming specialized models like Gemini-2.5-Flash (38.02) and GPT-5 (37.81). The strong performance of EDITREWARD on both Qwen and MiMo-VL backbones also confirms that our framework is robust and effectively scales with more powerful base models.

## 4.4 APPLICATION: EDITREWARD AS A REWARD

To demonstrate EDITREWARD's practical utility as a data supervisor, we conducted a data curation experiment designed to improve a state-of-the-art editing model. We employed our reward model to score the 46,000 examples in the ShareGPT-4o-Image dataset (Chen et al., 2025a), from which we selected high-quality subsets (Top 10K, 20K, and 30K) for comparative analysis. This curated dataset was then used to fine-tune the powerful Step1X-Edit model (Liu et al., 2025b). The computational cost for scoring the samples in this pool was minimal, requiring only average 2.61 GPU hours, demonstrating high efficiency (0.25 seconds/sample).

Table 4: Comprehensive comparison of state-of-the-art models on both the English and Chinese versions of the GEdit-Bench benchmark, across intersection and full test sets. Our model, significantly improve the base model Step1X-Edit, including sensitivity analysis for the EDITREWARD-curated subsets (Top 10K, 20K, and 30K). ↑ indicates higher the better. *-I means intersection set.

| Model | GEdit-Bench-EN-I ↑ | | | GEdit-Bench-EN ↑ | | | GEdit-Bench-CN-I ↑ | | | GEdit-Bench-CN ↑ | | |
|---|---|---|---|---|---|---|---|---|---|---|---|---|
| | G_SC | G_PQ | G_O | G_SC | G_PQ | G_O | G_SC | G_PQ | G_O | G_SC | G_PQ | G_O |
| AnyEdit (Yu et al., 2025) | 3.122 | 5.865 | 2.919 | 3.053 | 5.882 | 2.854 | 3.098 | 5.840 | 2.899 | 3.011 | 5.849 | 2.817 |
| OmniGen (Wu et al., 2025b) | 6.037 | 5.856 | 5.154 | 5.879 | 5.871 | 5.005 | 6.015 | 5.830 | 5.122 | 5.850 | 5.845 | 4.976 |
| Gemini-2.0 (Hassabis et al., 2024) | 6.816 | 7.408 | 6.483 | 6.866 | 7.436 | 6.509 | 6.790 | 7.385 | 6.450 | 6.821 | 7.402 | 6.473 |
| Doubao (Wang et al., 2025c) | 7.396 | 7.899 | 7.137 | 7.222 | 7.885 | 6.983 | 7.370 | 7.870 | 7.105 | 7.195 | 7.851 | 6.942 |
| GPT-Image-1 (OpenAI, 2025) | 7.867 | 8.097 | 7.590 | 7.743 | 8.133 | 7.494 | 7.840 | 8.075 | 7.560 | 7.708 | 8.095 | 7.451 |
| Step1X-Edit | 7.289 | 6.962 | 6.618 | 7.131 | 6.998 | 6.444 | 7.464 | 7.076 | 6.779 | 7.647 | 7.398 | 6.983 |
| Step1X-Edit + ShareGPT-4o-Image | 7.411 | 6.838 | 6.803 | 7.349 | 6.893 | 6.780 | 7.126 | 6.855 | 6.595 | 7.116 | 6.807 | 6.583 |
| **Ours (EDITREWARD as reward) (Top-K Sensitivity)** | | | | | | | | | | | | |
| Step1X-Edit + Ours (Top 10K) | 7.762 | 6.811 | 6.957 | 7.690 | 6.866 | 6.938 | 7.591 | 7.064 | 7.000 | 7.591 | 7.047 | 6.987 |
| Step1X-Edit + Ours (Top 30K) | 7.641 | 6.957 | 7.007 | 7.632 | 6.890 | 6.962 | 7.524 | 7.068 | 6.938 | 7.456 | 7.098 | 6.888 |
| **Step1X-Edit + Ours (Top 20K)** | **7.895** | **6.946** | **7.131** | **7.854** | **6.931** | **7.086** | **7.757** | **7.024** | **7.074** | **7.658** | **6.995** | **7.001** |

**Evaluation Protocol.** To measure the impact of this curation, we evaluate the resulting model on the comprehensive **GEdit-Bench**. This benchmark features both English (EN) and Chinese (CN) instructions, as well as a challenging "Intersection" subset containing prompts that all models could process. We compare our fine-tuned model against a diverse range of baselines, including the original Step1X-Edit, the same model fine-tuned on the *full* unfiltered dataset, and other leading open-source and proprietary models like Doubao and GPT-Image-1. Following established practices Ku et al. (2023), performance is judged by GPT-4o on three metrics (0-10 scale): **Semantic Consistency (G_SC)** for instruction fidelity, **Perceptual Quality (G_PQ)** for visual realism, and an **Overall Score (G_O)** for overall quality.

**Results and Analysis.** As detailed in Table 4, this reward-driven filtering yields significant performance gains. The results show a clear trade-off between data quality and volume, confirming that our EDITREWARD-filtered 20K subset represents the optimal balance for fine-tuning. Our best-performing model, trained on the Top 20K subset, achieves an English G_O score of **7.086**. This substantially outperforms the original Step1X-Edit baseline (6.444) and the model trained on the full, noisy 46K dataset (6.780). Furthermore, our sensitivity analysis confirms that while the Top 10K subset (representing the highest signal-to-noise ratio) also outperforms the full set (G_O: 6.938), it is marginally inferior to the Top 20K subset, indicating the 20K size is necessary for robust generalization and avoiding underfitting. Crucially, the Top 30K subset (G_O: 6.962) yields diminishing returns compared to the Top 20K, confirming that including lower-quality data dilutes the training signal. This finding is crucial, as it confirms that data quality, as judged by our reward model, is more impactful than sheer data quantity. EDITREWARD successfully prunes noisy examples that would otherwise degrade performance during fine-tuning. This uplift elevates the open-source Step1X-Edit to be competitive with top-tier editors like Doubao, validating our model's potential as an essential tool for training next-generation generative models.

## 4.5 OUT-OF-DISTRIBUTION GENERALIZATION ANALYSIS

To evaluate robustness outside the training pool, we conducted a targeted experiment on two challenging Out-of-Distribution (OOD) categories: **Text-in-Image** (OCR) and **Style Transfer**.

Table 5: Accuracy comparison on OOD tasks (Text & Style) sourced from Open Images. EDITRE-WARD achieves performance comparable to GPT-4o while being open-source and cost-effective.

| Model | Text Category | Style Category | Overall |
|---|---|---|---|
| GPT-4o | 45.50 | 35.79 | 41.69 |
| **EDITREWARD (on MiMo-VL-7B-SFT)** | **47.83** | **45.41** | **46.80** |

**Experimental Setup.** We constructed a specialized OOD set sourced from **Open Images** (distinct from training sources), comprising **253 Text pairs** and **185 Style pairs** with expert annotations. We compare EDITREWARD (on MiMo-VL-7B-SFT) against the commercial SOTA, **GPT-4o**.

**Results.** Table 5 shows EDITREWARD achieves performance comparable to GPT-4o on these tasks, maintaining competitive alignment despite inherent VLM difficulties with OCR. Crucially, EDITRE-WARD offers significant advantages as a cost-effective, open-source alternative with faster inference speeds.

## 4.6 ABLATION STUDIES

Table 6: Ablation study on key design choices for our reward model. We compare a point-wise regression loss (variant I) against our pair-wise uncertainty loss (variant II, III, IV, V), and further investigate the impact of the reward head architecture (Shared vs. Multiple) and different score aggregation strategies.

| Variants | Model Configuration | | | Benchmark Performance | | | |
|---|---|---|---|---|---|---|---|
| | Loss Type | Head Type | Aggregation | **GenAI-Bench** | **AURORA-Bench** | **ImagenHub** | **EditReward** |
| I | Point-wise | N/A | N/A | 49.62 | 42.38 | 13.40 | 22.73 |
| II | Pair-wise | Shared | Mean | 60.17 | 56.75 | 32.65 | 36.78 |
| III | Pair-wise | Multiple | Min | 59.96 | 57.25 | 30.25 | 36.57 |
| IV | Pair-wise | Multiple | Sum | 59.63 | 55.19 | 32.93 | **37.60** |
| V | Pair-wise | Multiple | Mean | **63.97** | **59.50** | **36.18** | 36.78 |

**Ablation on Model Design.** We analyze our model's key architectural choices in Table 6.

*Loss Type.* Comparing loss functions (Variant I vs. V), our pair-wise uncertainty model (63.97 on GenAI-Bench) significantly outperforms the point-wise regression baseline (49.62). This confirms that modeling relative preferences is more effective than regressing on absolute scores for this task.

*Head Type.* For the reward head (Variant II vs. V), using multiple independent heads (63.97) provides a clear improvement over a shared architecture (60.17 on GenAI-Bench), suggesting that specialized heads better capture our disentangled evaluation dimensions.

*Aggregation Strategy.* Finally, we compare three score aggregation strategies (Variants III-V), finding that the balanced mean provides the most consistent and highest performance (63.97 on GenAI-Bench and 59.50 on AURORA-Bench). We therefore adopt the **Pair-wise model with Multiple heads and Mean aggregation** as our final configuration.

Table 7: Ablation study on different model parameter sizes and different model backbones.

| Backbone | GenAI-Bench | AURORA-Bench | ImagenHub | EDITREWARD-BENCH |
|---|---|---|---|---|
| Qwen2.5-VL-3B-Inst | 62.79 | 57.37 | 32.34 | 37.40 |
| Qwen2.5-VL-7B-Inst | 63.97 | 59.50 | **36.18** | 36.78 |
| MiMo-VL-7B-SFT-2508 | **65.72** | **63.62** | 35.20 | **38.42** |

**Ablation on Model Backbone.** To verify our framework's generalizability, we train EDITREWARD on three backbones of varying scale and architecture, confirming that our method consistently benefits from stronger foundation models (Table 7). Performance increases when scaling from **Qwen2.5-VL-3B** to **7B**, and improves further at the 7B scale when using the more advanced **MiMo-VL-7B**

architecture, which achieves state-of-the-art scores of **65.72%** on GenAI-Bench and **63.62%** on AURORA-Bench. This demonstrates that our framework is backbone-agnostic and effectively leverages the capabilities of more powerful models.

## 5 RELATED WORKS

**Evolution of Instruction-Guided Image Editing.** Instruction-guided image editing has rapidly evolved from early trajectory-based methods. Diffusion models (Song et al., 2020; Dhariwal & Nichol, 2021; Rombach et al., 2022; Podell et al., 2023) first enabled editing via dual-prompt formulations that relied on cross-attention manipulation or inversion (Hertz et al., 2022; Mokady et al., 2023; Wallace et al., 2023). The paradigm then shifted to more user-friendly single-instruction editing, pioneered by InstructPix2Pix (Brooks et al., 2023) and refined by works like MagicBrush and Emu-Edit (Zhang et al., 2024a;b; Sheynin et al., 2024) that focused on curating high-quality datasets. This trajectory-based family has been further advanced by flow-matching models (BlackForestLabs et al., 2025), which improve training and sampling efficiency. In parallel, sequential generative models, including autoregressive approaches (Yu et al., 2022; Tian et al., 2024), enhance compositional reasoning. The most recent advances feature hybrid multimodal architectures like OmniGen2 (Wu et al., 2025b) and BAGEL (Deng et al., 2025), which integrate large vision–language backbones with generative decoders to enable more context-aware, conversational editing.

**Evaluating Instruction-Guided Image Editing.** Early evaluation of image editing relied on perceptual metrics like LPIPS (Zhang et al., 2018), but these require reference images and fail to assess semantic alignment. CLIP-based metrics (Hessel et al., 2021) were introduced for text–image consistency but also show limited correlation with human judgment (Ku et al., 2024). The advent of large vision–language models (VLMs) enabled zero-shot evaluation, with proprietary models (Ku et al., 2023; Wang et al., 2025b) demonstrating promising human correlation while open-source counterparts (Liu et al., 2023; Laurençon et al., 2024) have lagged (Jiang et al., 2024). Consequently, recent work has focused on improving open-source evaluators via fine-tuning. One strategy distills supervision from proprietary models (Wei et al., 2024; Gu et al., 2024; Wu et al., 2024), which risks inheriting model biases. The other collects direct human annotations (Xu et al., 2024; Liang et al., 2024; Wu et al., 2023; Sani et al., 2026), offering higher-quality signals but typically at a smaller scale. Our work contributes a large-scale, expert-annotated dataset, enabling more reliable and robust reward modeling for image editing.

## 6 CONCLUSION

In this paper, we addressed the critical bottleneck hindering the advancement of open-source instruction-guided image editing: the lack of a reliable, human-aligned reward model for scaling up high-quality training data. To this end, we introduced a three-part solution: (1) **EDITREWARD-DATA**, a new large-scale (200K) preference dataset curated with rigorous expert annotation to minimize the noise and bias prevalent in existing resources; (2) **EDITREWARD**, a dedicated reward model trained on this high-fidelity data to specialize in the image editing domain; and (3) **EDITREWARD-BENCH**, a challenging new benchmark featuring multi-way preference tasks to enable more robust evaluation. Our experimental results validate the effectiveness of our approach. EDITREWARD establishes a new state of the art, demonstrating superior correlation with human judgment by outperforming strong VLM judges like GPT-5 and GPT-4o on public benchmarks. More importantly, we demonstrated its practical utility in a downstream data curation task: fine-tuning Step1X-Edit on a 20K subset of data filtered by EDITREWARD yielded significantly better performance than training on the full 46K noisy dataset (7.1 vs. 6.7 overall score on GEdit-Bench). This confirms that a high-quality reward signal is a key ingredient for training powerful, next-generation editing models. Ultimately, this work provides both a methodology and a set of open resources to help bridge the gap between open-source and proprietary image editing models. To empower the community and facilitate future research, we will publicly release our EDITREWARD-DATA dataset, the trained EDITREWARD model, and the EDITREWARD-BENCH benchmark.

## ETHICS STATEMENT

The development of advanced instruction-guided image editing models, which our work aims to evaluate and improve, carries significant ethical implications. While these technologies enable powerful creative expression, they can also be misused to generate deceptive or harmful content, such as deepfakes, misinformation, or fraudulent documents, lowering the barrier for malicious actors. Our work, by creating a more effective reward model, could inadvertently contribute to accelerating these capabilities. We acknowledge this dual-use potential and have taken steps to mitigate risks. Specifically, the EDITREWARD-DATA dataset was constructed from publicly available, non-sensitive benchmarks, and automated and manual filtering was applied to remove any personally identifiable information (PII) or sensitive content. Our reward model, EDITREWARD, is trained to align with constructive and high-quality edits, as defined by our multi-dimensional rubric, and does not follow harmful or malicious instructions. Additionally, all generated data and model outputs will be released under a CC-BY-NC-SA 4.0 license, explicitly prohibiting commercial use, which mitigates potential misuse such as the creation of deepfakes or other harmful applications. By publicly releasing our dataset, model, and code, we aim to promote transparency and enable the research community to further study the safety, biases, and alignment of such models. Finally, we encourage the community to adopt similar safeguards, including watermarking, provenance tracking, and careful curation of training data, when deploying or extending instruction-guided image editing models.

## REPRODUCIBILITY STATEMENT

To ensure the reproducibility of our work, we provide the following details. All of our reward models were trained on 8 NVIDIA A800 GPUs. The evaluation of baseline models was conducted using their official public codebases and recommended configurations. For proprietary models (e.g., GPT-4o, Gemini series), we accessed their APIs between April and June 2025; given the evolving nature of these models, we have archived their specific outputs for consistency. Our new dataset, EDITREWARD-DATA, was constructed following the detailed protocol described in Section 2.1, and both the dataset and our evaluation benchmark, EDITREWARD-BENCH, will be publicly released. The complete codebase for training and evaluating our EDITREWARD, along with the final model weights for both the Qwen2.5-VL and MiMo-VL backbones, will be made available on GitHub and Hugging Face. Further details are provided in Appendix A.4 and A.5.

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

# A APPENDIX

## A.1 USE OF LLM

Large Language Models (LLMs) were used exclusively for minor grammar correction and stylistic refinement of the manuscript. Their role was purely auxiliary, and all major scientific contributions were made by the authors. The authors bear full responsibility for the content of this work.

## A.2 DETAILS OF EDITREWARD-DATA CONSTRUCTION

Our dataset construction was centered on three principles: **ecological validity**, by sourcing instructions from human-vetted benchmarks; **diversity**, by generating candidates from state-of-the-art models; and **reliability**, through a rigorous multi-dimensional annotation pipeline.

**Source Data Collection.** To ensure ecological validity, we collected 9,557 unique instruction-image pairs from six established, human-vetted sources: GEdit-Bench (606), ImgEdit-Bench (737), MagicBrush (1,053), AnyEdit (1,250), EmuEdit (5,611), and a challenging internal set (300). This aggregation provides a comprehensive foundation of semantically grounded and executable edit instructions across a wide spectrum of tasks and styles.

**Candidate Generation.** For each of the 9,557 source pairs, we generated a diverse pool of 12 candidate images using six state-of-the-art models: **Step1X-Edit** (Liu et al., 2025b), **Flux-Kontext** (BlackForestLabs et al., 2025), Qwen-Image-Edit (Wu et al., 2025a), **BAGEL** (Deng et al., 2025), **Ovis-U1** (Wang et al., 2025a), and **OmniGen2** (Wu et al., 2025b). To ensure a broad quality spectrum and mitigate model-specific biases, we utilized multiple random seeds, preventing any single model from dominating the candidate pool.

Table 8: The detailed comparison of different generative preference datasets and benchmarks.

| Dataset | Venue | Scale | Task Focus | Annotation | Eval. Dims. | Limitation / Caveat |
|---|---|---|---|---|---|---|
| ImageRewardDB | NeurIPS'23 | ~137K | Visual Generation | Human | Single | Expert comparisons with limited variety |
| VisionPrefer | NeurIPS'24 | 1.2M | Generation | Model | Multiple | Multi-aspect but model-derived bias risks |
| GenAI-Bench | NeurIPS'24 | ~1.6K | Generation / Editing | Human | Multiple | High quality but very small scale |
| HIVE | CVPR'24 | ~3.6K | Instructional Editing | Human | Single | Task-specific, limited comparison set size |
| ADIEE | ICCV'25 | >100K | Instructional Editing | Model | Single | Synthetic labels; possible model bias |
| HPDv3 | ICCV'25 | >1.17M | Visual Generation | Human | Single | Wide-spectrum; generalizability limits |
| **EDITREWARD-DATA** | | ~200K | Instructional Editing | Human | Multiple | Large scale and fine-grained supervision |

| Benchmark | Venue | Scale | Annotation | Eval. Dims. | Multi-Way Preference | Pair-Wise | Point-Wise |
|---|---|---|---|---|---|---|---|
| GenAI-Bench | NeurIPS'24 | ~900 | Human | Multiple | 2-way | ✓ | — |
| AURORA-Bench | NeurIPS'24 | ~1.6K | Human | Multiple | 2-way | ✓ | ✓ |
| ImagenHub | ICLR'24 | ~1.4K | Human + Model | Single | 2-way | — | ✓ |
| **EDITREWARD-BENCH** | | 500 Groups (~1.5K) | 3 Human (Cross-check) | Multiple | 2/3/4-way | ✓ | ✓ |

**Multi-Dimensional Annotation.** From the pool of 12 candidates, 7 were randomly sampled for human evaluation. Annotators provided two separate scores for each candidate on a 4-point Likert scale (1=Poor to 4=Excellent), corresponding to our two evaluation dimensions: (1) **Instruction Following**, which assesses semantic accuracy, completeness, and the avoidance of unprompted changes; and (2) **Visual Quality**, which evaluates physical plausibility, absence of artifacts, and overall aesthetic appeal. This multi-dimensional rubric provides a more granular assessment than a single holistic score. Detailed interface of the annotations is in Figure 4. We also provide detailed annotation guidance below.

**Annotation Guidelines:**

### Instruction Following

*This dimension focuses on how accurately, completely, and exclusively the model executed the text instruction.*

**Key Criteria:**
- **Semantic Accuracy:** Correctly interpreting the core meaning.
- **Completeness:** Fulfilling all parts of the instruction.
- **Exclusivity:** Avoiding unprompted changes to the rest of the image.

**Negative Indicators:**
- A key part of the instruction is ignored (e.g., color changed but not the object).
- A major misinterpretation (e.g., "orange" yields a grapefruit).
- The image is unchanged or a random, unrelated image is generated.

**Scoring Rubric (1-4 Scale):**
- **4 (Very Good):** Perfectly executes all aspects of the instruction. Edit is surgical and flawless.
- **3 (Relatively Good):** Achieves the main goal but with minor deviations or omissions (e.g., misses a small detail).
- **2 (Relatively Poor):** Significantly misunderstands or only partially executes the instruction. Unedited areas may be noticeably altered.
- **1 (Very Poor):** Completely fails the instruction. The result is unrelated, or the image is corrupted.

### Visual Quality

*This dimension focuses on the physical plausibility, technical flawlessness, and overall aesthetic appeal of the edited image.*

**Key Criteria:**
- **Plausibility:** Consistency with real-world physics (lighting, shadows).
- **Artifact-Free:** Absence of visual flaws (blur, distortion, seams).
- **Aesthetic Quality:** The overall harmony, naturalness, and visual appeal.

**Negative Indicators:**
- Obvious physical errors (e.g., an object casts no shadow).
- Noticeable and distracting artifacts (e.g., a blurry halo around the edit).
- The final image is jarring, ugly, or unbalanced.

**Scoring Rubric (1-4 Scale):**
- **4 (Very Good):** Perfectly realistic and visually flawless. The edit is undetectable and appealing.
- **3 (Relatively Good):** High quality overall, but close inspection may reveal minor imperfections (e.g., shadow is slightly off).
- **2 (Relatively Poor):** The edit is obvious and looks unnatural, with clear visual flaws that detract from its quality.
- **1 (Very Poor):** A visual failure, full of severe errors and artifacts, making it unusable.

**Quality Control.** The reliability of our annotations is ensured through a multi-stage process. The process includes: (1) initial pilot studies to refine the annotation guidelines and rubric; (2) a formal training and calibration phase for all annotators to align their judgments; and (3) continuous random sampling and cross-checking of annotations during the formal labeling process to maintain a

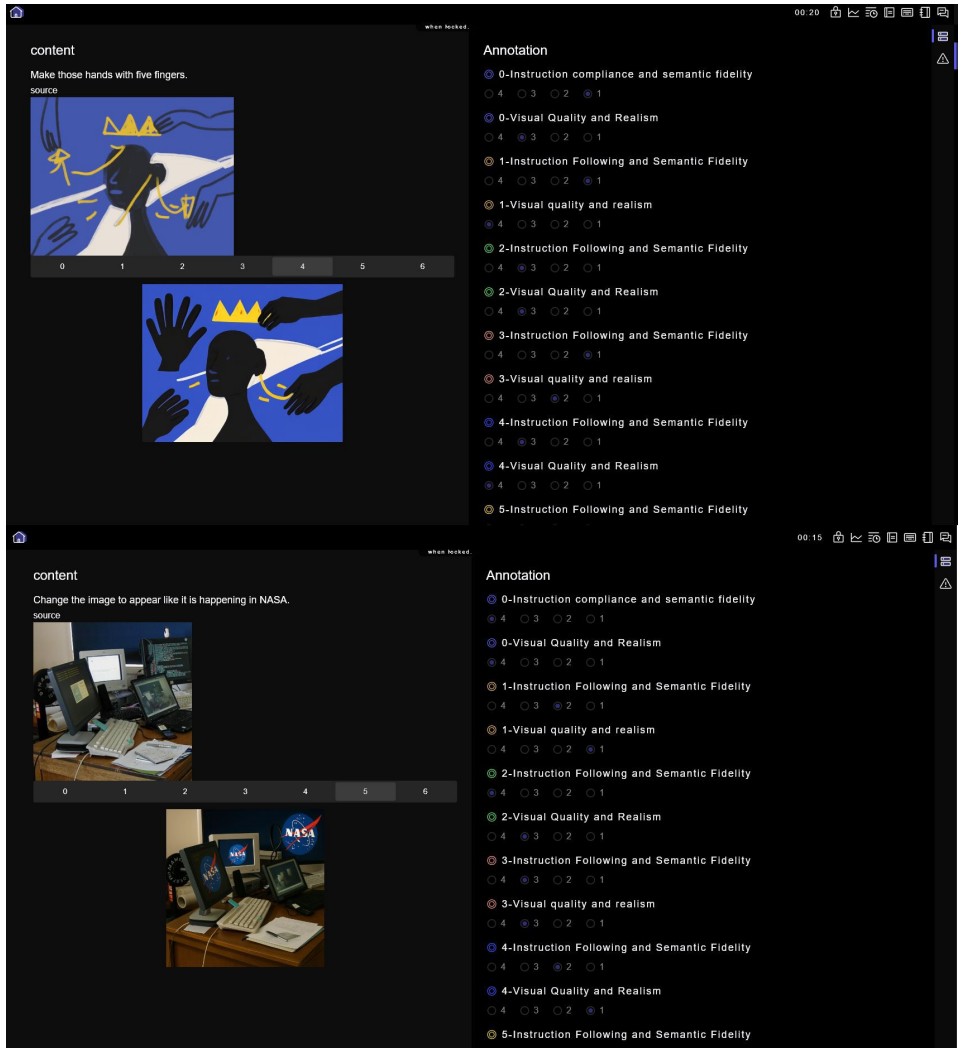

Figure 4: Annotation Interface

Table 9: Inter-Annotator Agreement (IAA) Metrics for Expert Annotations. The robust Krippendorff Alpha ($\alpha$) values confirm the high reliability of our multi-dimensional expert scoring.

| Dataset | Fleiss' Kappa (IF) | Fleiss' Kappa (VQ) | Krippendorff Alpha (IF) | Krippendorff Alpha (VQ) |
|---|---|---|---|---|
| EDITREWARD-DATA | 0.4157 | 0.3203 | 0.6762 | 0.5720 |
| EDITREWARD-BENCH | 0.3962 | 0.3157 | 0.6623 | 0.6114 |
| All data | 0.3994 | 0.3111 | 0.6685 | 0.5972 |

high inter-annotator agreement (IAA). More representative examples from EDITREWARD-DATA are shown in the Appendix A.10. The IAA results in Table 9 provide a critical quantitative assessment of our expert annotation quality (Fleiss, 1971). We highlight the values derived from Krippendorff's Alpha ($\alpha$) (Krippendorff, 2011), which is the most appropriate metric as it correctly models the ordinal nature of our 4-point Likert scale. The $\alpha$ scores of **0.668** for Instruction Following (IF) and **0.597** for Visual Quality (VQ) establish a strong, quantified baseline for human consistency. Crucially, the observed difference ($\text{IAA}_{\text{IF}} > \text{IAA}_{\text{VQ}}$) provides empirical validation for our core contribution: it confirms that the VQ dimension is inherently more subjective than IF. This validates our design choice to use a multi-dimensional rubric and a multi-head reward model, as a single holistic score would obscure this critical difference in human variance.

## A.3  DETAILS OF EDITREWARD-BENCH CONSTRUCTION

To provide a more robust and discerning evaluation of image editing reward models, we introduce EDITREWARD-BENCH. The design of this new benchmark is motivated by several limitations identified in existing evaluation suites. For instance, **ImagenHub** utilizes a simple 3-point rating scale [0, 0.5, 1]. While user-friendly, this coarse granularity can fail to capture the nuanced quality differences across the broad spectrum of semantic consistency and perceptual quality (Ku et al., 2023). The editing tasks in **AURORA-Bench** are primarily focused on action-centric and reasoning-centric instructions, which may not represent the full diversity of common editing requests.

To address these challenges, we constructed EDITREWARD-BENCH through a meticulous pipeline. The foundation of our benchmark is a curated subset of **500 high-quality groups** sampled from our EDITREWARD-DATA candidate pool, spanning 7 distinct editing categories. To establish a reliable ground truth, we engaged three independent groups of trained expert annotators. Following the multi-dimensional rubric detailed in Section 2.1, each annotator assigned scores on a 4-point Likert scale [1, 2, 3, 4] for both instruction fidelity and visual quality. This process ensures the robustness and accuracy of our ground-truth labels. To increase the benchmark's difficulty and test the fine-grained discriminative power of models, we prioritized the inclusion of samples where the competing edits have small differences in their average human scores.

The primary innovation of EDITREWARD-BENCH is its introduction of a **multi-way preference comparison** protocol, moving beyond simple pairwise judgments. We construct more complex evaluation units, including **ternary tuples (A, B, C)** and **quaternary tuples (A, B, C, D)**, based on our reliable human scores. For a model's evaluation of a tuple to be considered correct, it must correctly predict the preference relationship for *all constituent pairs* within that tuple (e.g., A>B, A>C, and B>C for a ternary tuple where A is the best and C is the worst). This strict, all-or-nothing criterion provides a much more comprehensive and robust measure of a reward model's ranking consistency and reasoning capabilities than traditional pairwise accuracy. We evaluated a wide range of leading models on EDITREWARD-BENCH to establish its utility. The experimental results are detailed in Section 4.

**Dataset Details** We provide additional details regarding our annotation protocol. All annotators followed a standardized rubric with clear dimension-specific guidelines, covering Instruction Following (IF) and Visual Quality (VQ). To ensure high consistency, each annotator underwent training sessions with reference examples before formal labeling.

For EDITREWARD-DATA, each edited image is scored by a single expert annotator on a 4-point scale (1–4) across the two dimensions (IF, VQ). This provides large-scale but fine-grained supervision.

For EDITREWARD-BENCH, every group is annotated by three independent experts, again along the two dimensions (IF, VQ). Annotators must jointly determine the ranking consistency among multiple candidates. When disagreements occur, a cross-check protocol ensures consistency across annotators, with the final label derived from majority agreement.

This protocol guarantees both the scale and quality of the training data and the strict reliability of the benchmark.

## A.4  MORE DETAILS AND IMPLEMENTATION OF TRAINING

**Reward Model Architecture.** Our reward model, EDITREWARD, is built upon a powerful pre-trained Vision-Language Model (VLM) backbone, which is fully fine-tuned during training. Our main results use two backbones: **Qwen2.5-VL-7B** and **MiMo-VL-7B**. The VLM backbone is followed by a Multi-Layer Perceptron (MLP) reward head. Based on our ablation studies, we use a **Multiple Head** architecture, where separate MLP heads predict the parameters $(\mu, \sigma^2)$ for each of the two quality dimensions independently.

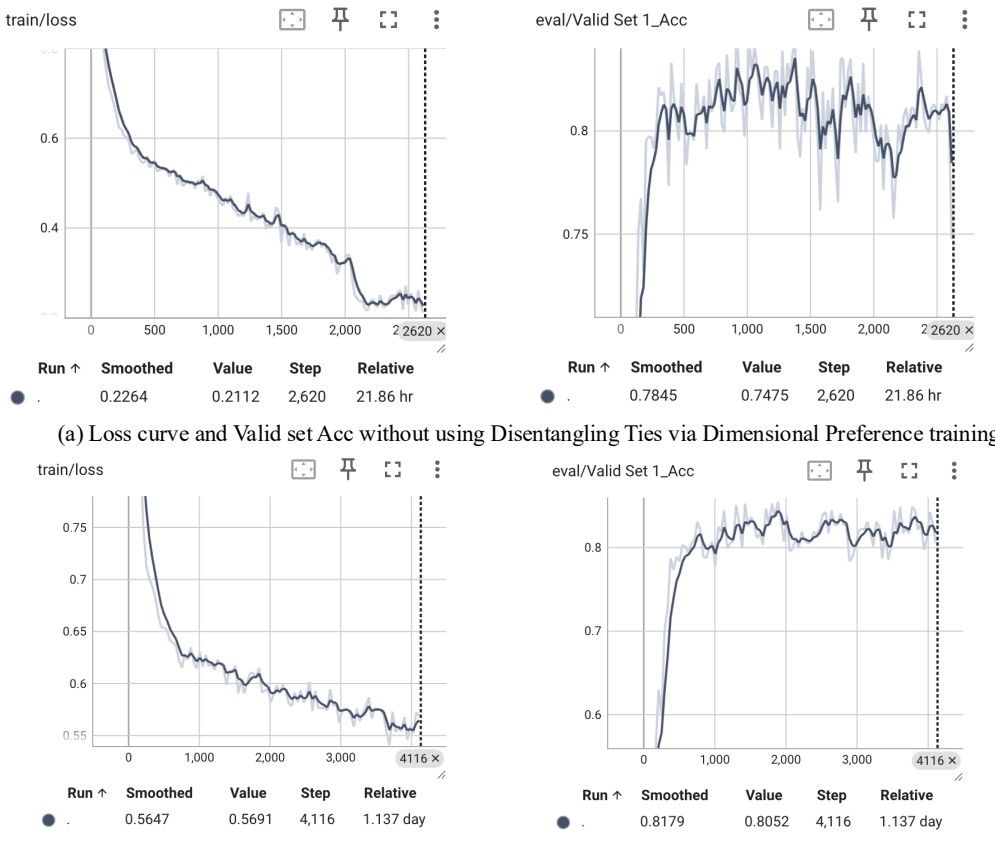

(a) Loss curve and Valid set Acc without using Disentangling Ties via Dimensional Preference training

(b) Loss curve and Valid set Acc by using Disentangling Ties via Dimensional Preference training

Figure 5: Loss curve and Valid set Acc by using or not using Disentangling Ties via Dimensional Preference during model training.

## A.5    MORE DETAILS ABOUT EVALUATION

We present the main experimental results in Table 3. The findings clearly demonstrate that our reward model, EDITREWARD, sets a new state of the art in aligning with human preferences for instruction-guided image editing.

**State-of-the-Art Performance.**  Our best model, **EDITREWARD (on MiMo-VL-7B)**, achieves the highest performance on three out of four benchmarks. It obtains a state-of-the-art accuracy of **65.72%** on GenAI-Bench, significantly surpassing the strongest proprietary competitor, GPT-5 (59.61), and the best open-source VLM, ADIEE (59.96). Similarly, on AURORA-Bench, our model scores **63.62%**, demonstrating a substantial margin over the next-best models, EDITREWARD (on Qwen2.5-VL-7B) at 59.50% and ADIEE at 55.56%. On ImagenHub, our models remain highly competitive with the top proprietary systems, with EDITREWARD (on Qwen2.5-VL-7B) achieving a Spearman correlation of 36.18, second only to GPT-4o.

**Effectiveness of Reward Modeling.**  A key insight from our results is the profound impact of our reward modeling framework itself. By comparing the base open-source VLMs to our EDITREWARD trained on them, we can quantify the performance uplift. For instance, the base **Qwen2.5-VL-7B-Inst** scores 40.48% on GenAI-Bench. After being trained with our multi-dimensional, uncertainty-aware methodology, the resulting **EDITREWARD (on Qwen2.5-VL-7B)** skyrockets to 63.97%—a massive **+23.5 point** improvement. This demonstrates that our contribution is not merely the application of a strong backbone, but a highly effective training paradigm that dramatically enhances a model's alignment with human judgments.

**Performance on EDITREWARD-BENCH and Backbone Generalization.**  Our proposed benchmark, EDITREWARD-BENCH, proves to be a more challenging and discerning testbed. Here, our

EDITREWARD (on MiMo-VL-7B) again achieves the top score of **38.42%**, narrowly outperforming Gemini-2.5-Flash (38.02) and GPT-5 (37.81). Notably, GPT-4o, the best-performing model on ImagenHub, scores significantly lower at 28.31, confirming that EDITREWARD-BENCH effectively identifies limitations in models that other benchmarks may miss. Finally, the strong performance of EDITREWARD on both Qwen2.5-VL and the more powerful MiMo-VL backbone confirms that our reward modeling framework is robust and can effectively leverage the capabilities of stronger base models to push the state of the art even further.

## A.6 MORE DETAILS ABOUT APPLICATION

Beyond direct evaluation, a key application for a powerful reward model is to improve downstream generative models through data curation. To demonstrate the practical utility of EDITREWARD, we conducted an experiment to see if it could filter a large, noisy dataset to create a high-quality subset for fine-tuning a state-of-the-art image editing model.

**Experimental Setup.** Our experiment uses the open-source **Step1X-Edit** (Liu et al., 2025b) as the base model for fine-tuning. The training data is derived from **ShareGPT-4o-Image** (Chen et al., 2025a), a large dataset containing approximately 46,000 instruction-image pairs. We first employed EDITREWARD to score every example in this dataset. We then curated a high-quality subset by selecting only the top-scoring 20,000 examples. The goal is to evaluate if fine-tuning Step1X-Edit on this smaller, curated subset yields better performance than training on the full, noisy dataset.

**Evaluation Metrics and Baselines.** We evaluate all models on **GEdit-Bench**, a comprehensive benchmark with English (EN) and Chinese (CN) versions, each containing a full set and a more challenging intersection (-I) split. Performance is measured across three axes: **Semantic Consistency (G_SC)**, which evaluates how well the edit follows the instruction; **Perceptual Quality (G_PQ)**, which assesses visual realism and aesthetics; and a holistic **General Overall (G_O)** score. For all metrics, higher is better.

We compare our final model against two critical baselines to measure the impact of our data curation:

- **Step1X-Edit**: The original model without any additional fine-tuning.
- **Step1X-Edit + ShareGPT-4o-Image**: The baseline model fine-tuned on the *full*, unfiltered ShareGPT-4o-Image dataset.

This setup allows us to directly isolate the benefit of filtering with EDITREWARD. We also compare against other leading editing models like Doubao and GPT-Image-1 to contextualize our performance.

**Results and Analysis.** As shown in Table 4, fine-tuning Step1X-Edit on our EDITREWARD-curated subset yields substantial improvements across all benchmarks and metrics. On the GEdit-Bench-EN Overall score (G_O), our model achieves **7.086**, a significant gain over both the original Step1X-Edit (6.444) and the model trained on the full, noisy dataset (6.780).

This result is crucial: it demonstrates that training on a smaller, higher-quality dataset curated by our reward model is more effective than training on the entire noisy dataset. EDITREWARD successfully identifies and filters out low-quality or misaligned examples that can harm the fine-tuning process. Furthermore, this improvement elevates the performance of the open-source Step1X-Edit to be on par with, or even superior to, strong competitors like Doubao (6.983). This experiment validates the high potential of EDITREWARD as an essential tool for data curation in the training pipelines of next-generation image editing models. In Figure x, we show how our reward model is used to score some image editing examples.

## A.7 MORE ABLATION EXPERIMENTS RESULTS

**Ablation on Data Scale and Tie Disentanglement.** Next, we investigate the combined effect of increasing our training data from 130k to 200k samples while also applying our proposed tie-disentanglement strategy. The results of this significant upgrade are presented in Table 10. Comparing our baseline model (Variant I) against our final model which incorporates both changes (Variant

Table 10: Ablation study on dataset size and our tie-disentanglement strategy.

| Variants | Ablation Setting | | Benchmark Performance | | | |
|---|---|---|---|---|---|---|
| | Dataset Size | Disentangling Ties | GenAI-Bench | AURORA-Bench (Pair) | ImagenHub | EDITREWARD-BENCH |
| *Direct ablation on the full dataset* | | | | | | |
| I | 130k | | 62.24 | 51.36 | 32.45 | 37.81 |
| II | 200k | ✓ | 63.97 | 53.33 | 36.18 | 36.78 |

II), we observe consistent performance gains across all public benchmarks. The improvement is most pronounced on ImagenHub, where the score increases substantially from 32.45 to 36.18. We also see notable gains on GenAI-Bench (62.24 → 63.97) and AURORA-Bench (51.36 → 53.33). Interestingly, we note a slight performance decrease on our proposed EDITREWARD-BENCH, suggesting it may have different sensitivities to the data distribution. Overall, these results confirm the significant benefit of our full data strategy, which combines a larger, high-quality dataset with our novel technique for leveraging ambiguous tie pairs.

Table 11: Bias sensitivity analysis of Gemini 2.0 Flash under left/right bias conditions on **GenAI-Bench**.

| Condition | Accuracy (%) |
|---|---|
| Left Bias | 55.28 |
| Right Bias | 50.16 |
| Bias Sensitivity (Gap) | 5.11 |

## A.8 POSITIONAL BIAS

In the course of our evaluation on **GenAI-Bench**, we identified a notable case of **bias sensitivity** in the *Gemini 2.0 Flash* model when subjected to systematic position bias. Specifically, when we artificially manipulated the ground-truth labels to favor either left-side (A>B) or right-side (B>A) preferences—while correspondingly swapping the image positions to maintain correctness—we observed a consistent performance discrepancy. As shown in Table 11, the model achieved 55.28% accuracy under the left-bias condition but only 50.16% under the right-bias condition, yielding a 5.11% gap. This systematic difference indicates that the model exhibits a positional preference for left-side comparisons, which could distort evaluation outcomes if left unaddressed. To prevent such bias from affecting comparative results, **GenAI-Bench** adopts a randomized positioning strategy that shuffles the order of candidate images (A and B) for each comparison task. This ensures that evaluation outcomes are driven by genuine quality judgments rather than positional artifacts, thereby preserving fairness, robustness, and reliability across diverse model architectures.

## A.9 INPUT TEMPLATE FOR REWARD MODEL

This section provides the exact input prompt template used in all experiments to guide our reward model, EDITREWARD, in scoring the quality of an image edit.

---

**INSTRUCTION EDIT FOLLOWING TEMPLATE**

`[IMAGE]` You are tasked with evaluating an edited image **in comparison with the original source image** based on **Visual Quality & Realism**, and assigning a score from 1 to 4, with 1 being the worst and 4 being the best. This dimension focuses on how realistic, artifact-free, and aesthetically appealing the edited image is, while remaining consistent with the source image.

**Inputs Provided:**
- Source Image (before editing)
- Edited Image (after applying the instruction)
- Text Instruction

**Sub-Dimensions to Evaluate:**
- **Semantic Accuracy:** Assess whether the edited content accurately captures the semantics of the instruction. The edited result should precisely match the intended meaning. For example, if the instruction is "replace apples with oranges," the object must clearly be oranges, not other fruits.
- **Completeness of Editing:** Check whether **all parts** of the instruction are fully executed. For multi-step edits (e.g., "replace a red car with a blue bicycle"), both the color change and the object replacement must be done without omissions.
- **Exclusivity of Edit (No Over-Editing):** Ensure that only the requested parts are changed. The rest of the image (as seen in the source) should remain unaltered. For example, if the instruction only involves replacing an object, the background, lighting, and unrelated objects should not be unnecessarily modified.
**Scoring Criteria:**
- **4 (Very Good):** Perfectly accurate, complete, and exclusive execution of the instruction.
- **3 (Relatively Good):** Largely correct, but with minor omissions or slight over-editing.
- **2 (Relatively Poor):** Major misinterpretation, incomplete edits, or noticeable unintended changes.
- **1 (Very Poor):** Instruction ignored or completely wrong execution.

Text instruction – {text_prompt}

---

## INSTRUCTION EDIT QUALITY TEMPLATE

**[IMAGE]** You are tasked with evaluating an edited image **in comparison with the original source image** based on **Visual Quality & Realism**, and assigning a score from 1 to 4, with 1 being the worst and 4 being the best. This dimension focuses on how realistic, artifact-free, and aesthetically appealing the edited image is, while remaining consistent with the source image.

**Inputs Provided:**
- Source Image (before editing)
- Edited Image (after applying the instruction)
- Text Instruction

**Sub-Dimensions to Evaluate:**
- **Plausibility & Physical Consistency:** Check whether the edit aligns with the laws of physics and the scene context. Lighting, shadows, reflections, perspective, size, and interactions with the environment should all appear natural compared to the source image.
- **Artifact-Free Quality:** Look for technical flaws such as blur, distortions, pixel misalignment, unnatural textures, or seams around edited regions. High-quality results should be free from such visible artifacts.
- **Aesthetic Quality:** Evaluate the overall harmony and visual appeal. The image should look natural, balanced, and pleasant. Colors, composition, and atmosphere should enhance the image rather than degrade it.
**Scoring Criteria:**
- **4 (Very Good):** Perfectly realistic, artifact-free, seamless, and aesthetically pleasing.
- **3 (Relatively Good):** Mostly realistic and clean, with only minor flaws that do not significantly distract.
- **2 (Relatively Poor):** Noticeable physical inconsistencies or visible artifacts that make the edit unnatural.
- **1 (Very Poor):** Severe artifacts, incoherent composition, or visually unusable result.

Text instruction – {text_prompt}

---

**Full Input Template**

`[IMAGE]` You are tasked with evaluating an edited image **in comparison with the original source image**, and assigning a score from 1 to 8, with 1 being the worst and 8 being the best. This score should reflect **both how accurately the instruction was followed and the visual quality of the edited image**.

**Inputs Provided:**
- Source Image (before editing)
- Edited Image (after applying the instruction)
- Text Instruction

**Dimension 1: Instruction Following & Semantic Fidelity**
Evaluate how well the edited image follows the given instruction. Consider the following sub-dimensions:
- **Semantic Accuracy:** Check if the edited content accurately captures the intended meaning of the instruction. For example, if the instruction is "replace apples with oranges," the object must clearly be oranges, not other fruits.
- **Completeness of Editing:** Verify that all aspects of the instruction are fully executed. Multi-step edits should be completely applied without omissions.
- **Exclusivity of Edit (No Over-Editing):** Ensure that only the requested changes are applied; the rest of the image should remain consistent with the source image without unintended modifications.

**Dimension 2: Visual Quality & Realism**
Evaluate the realism, technical quality, and aesthetic appeal of the edited image. Consider the following sub-dimensions:
- **Plausibility & Physical Consistency:** Check whether the edit aligns with natural laws and scene context (lighting, shadows, reflections, perspective, and object interactions).
- **Artifact-Free Quality:** Assess for technical flaws such as blur, distortions, pixel misalignment, unnatural textures, or seams around edited regions.
- **Aesthetic Quality:** Consider overall harmony and visual appeal. Colors, composition, atmosphere, and balance should enhance the image without degrading realism.

**Scoring Criteria (1–8):**
- **8 (Very Good):** Perfect instruction following and flawless visual quality; edits are accurate, complete, exclusive, and visually seamless.
- **7 (Relatively Good):** Very good instruction following and high visual quality; minor, non-distracting flaws.
- **6 (Good):** Good instruction following or mostly good visual quality; minor omissions or slight artifacts.
- **5 (Moderate):** Partially correct edits or moderate visual issues; noticeable flaws but understandable.
- **4 (Relatively Poor):** Significant misinterpretation, incomplete edits, or noticeable visual artifacts.
- **3 (Poor):** Major errors in instruction following and/or poor visual quality; hard to fully understand.
- **2 (Very Poor):** Very poor edits with large semantic errors and strong visual artifacts.
- **1 (Failed):** Completely wrong edits or visually unusable result.

Text instruction – `{text_prompt}`

---

## A.10 REPRESENTATIVE RESULTS OF EDITREWARD-DATA

The following examples provide additional qualitative illustrations of EDITREWARD-DATA. They highlight a broad spectrum of real editing behaviors, including appearance manipulation, object insertion/removal, style transfer and text change. Each example includes the source image, the edited result, and the associated annotations—such as instruction-following and visual quality. These samples complement the main paper by demonstrating the dataset's diversity, annotation fidelity, and coverage across both everyday and challenging editing scenarios.

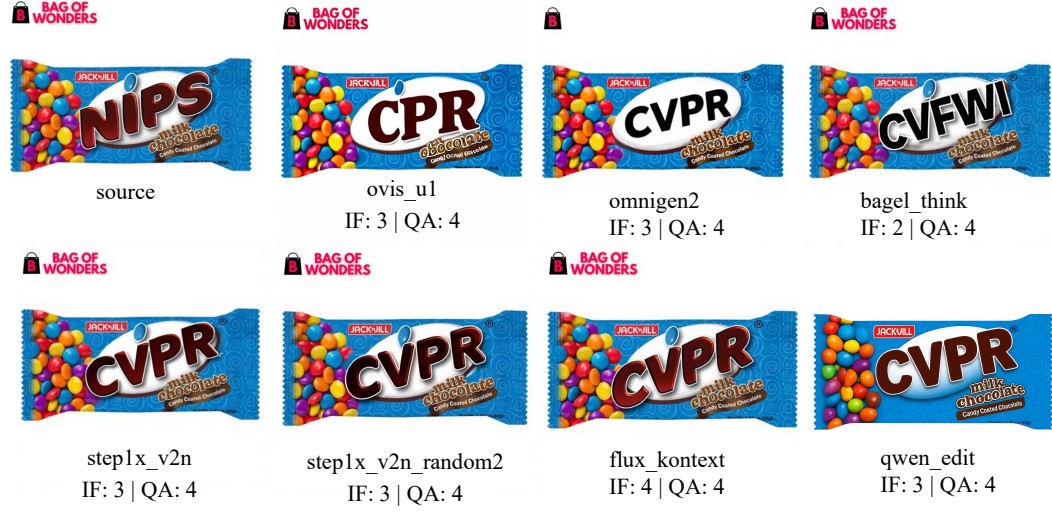

Instruction: Replace the text 'NIPS' with 'CVPR'

(a) Example 1 from EDITREWARD-DATA.

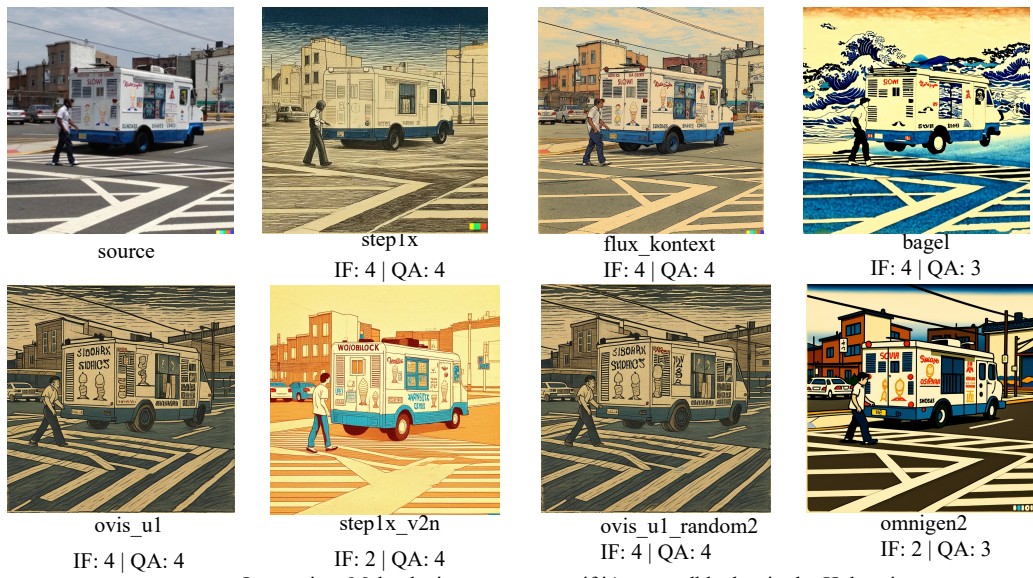

Instruction: Make the image appear as if it's a woodblock print by Hokusai.

(b) Example 2 from EDITREWARD-DATA.

Figure 6: Representative examples from EDITREWARD-DATA, complementing Fig. 2.

## A.11 QUALITATIVE EXAMPLES OF EDITREWARD-BASED FILTERING

To provide additional intuition about the preferences learned by EDITREWARD-DATA 's reward model, we visualize samples from the ShareGPT-4o-Image dataset (Chen et al., 2025a) that are either retained or filtered out after ranking with EditReward scores. The selected examples highlight the characteristic patterns captured by the reward model.

**High-Quality Retained Data.** Samples with high EditReward scores generally demonstrate accurate instruction following, clean and localized modifications, and visually coherent integration with the surrounding context. These images exhibit minimal artifacts and adhere closely to both the semantic intent and spatial constraints of the edit.

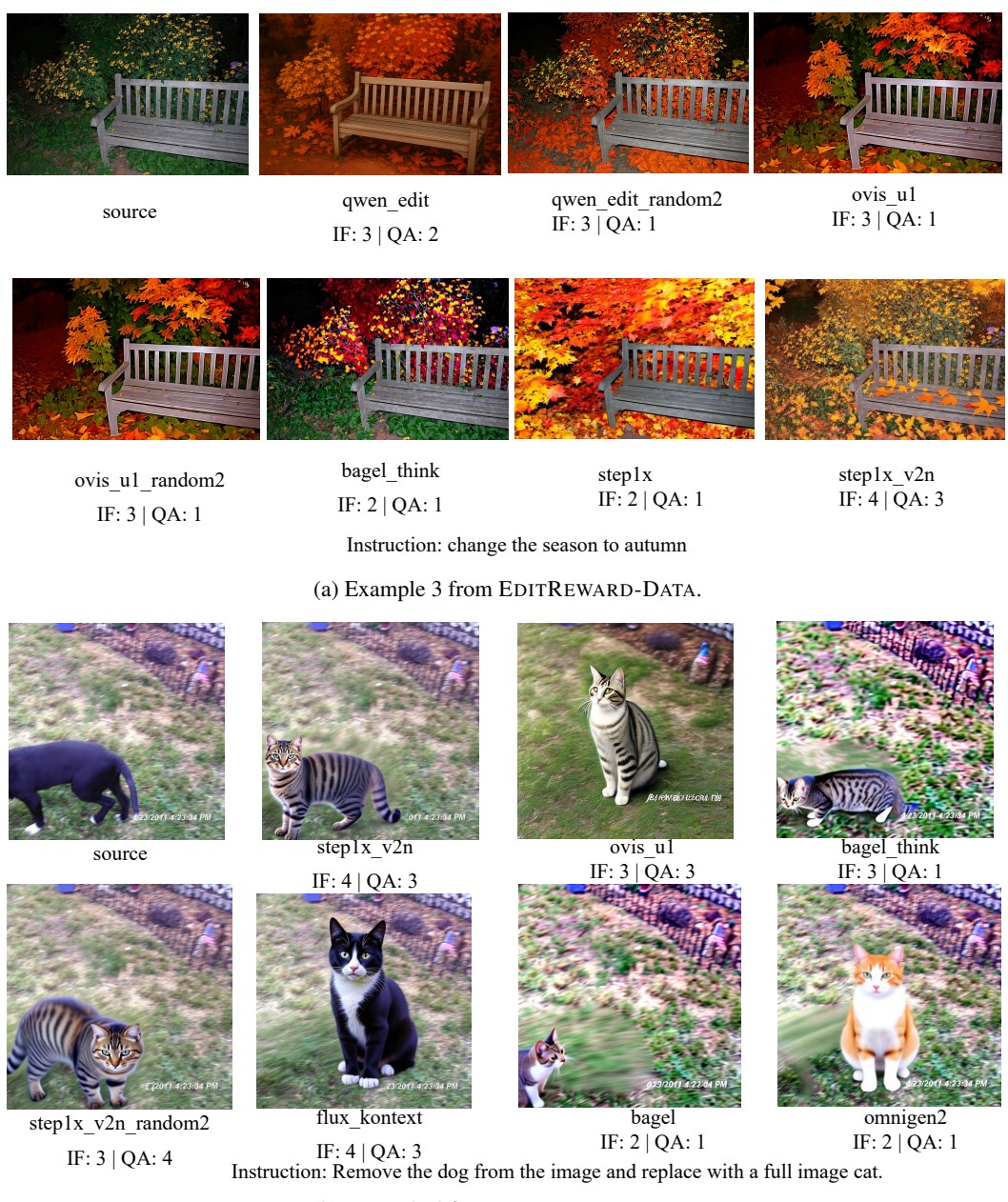

source

qwen_edit
IF: 3 | QA: 2

qwen_edit_random2
IF: 3 | QA: 1

ovis_u1
IF: 3 | QA: 1

ovis_u1_random2
IF: 3 | QA: 1

bagel_think
IF: 2 | QA: 1

step1x
IF: 2 | QA: 1

step1x_v2n
IF: 4 | QA: 3

Instruction: change the season to autumn

(a) Example 3 from EDITREWARD-DATA.

source

step1x_v2n
IF: 4 | QA: 3

ovis_u1
IF: 3 | QA: 3

bagel_think
IF: 3 | QA: 1

step1x_v2n_random2
IF: 3 | QA: 4

flux_kontext
IF: 4 | QA: 3

bagel
IF: 2 | QA: 1

omnigen2
IF: 2 | QA: 1

Instruction: Remove the dog from the image and replace with a full image cat.

(b) Example 4 from EDITREWARD-DATA.

Figure 7: Representative examples from EDITREWARD-DATA, complementing Fig. 2.

**Low-Quality Filtered Data.** Samples with low scores often contain undesirable visual artifacts, incorrect or incompletely executed edits, spatial misalignment, or hallucinated content. These failure patterns reflect typical challenges in image editing that violate instruction-following or degrade overall image quality.

Together, these qualitative examples illustrate the types of editing behaviors favored or penalized by EditReward, offering a clear and interpretable view of the model's learned preferences during data filtering.

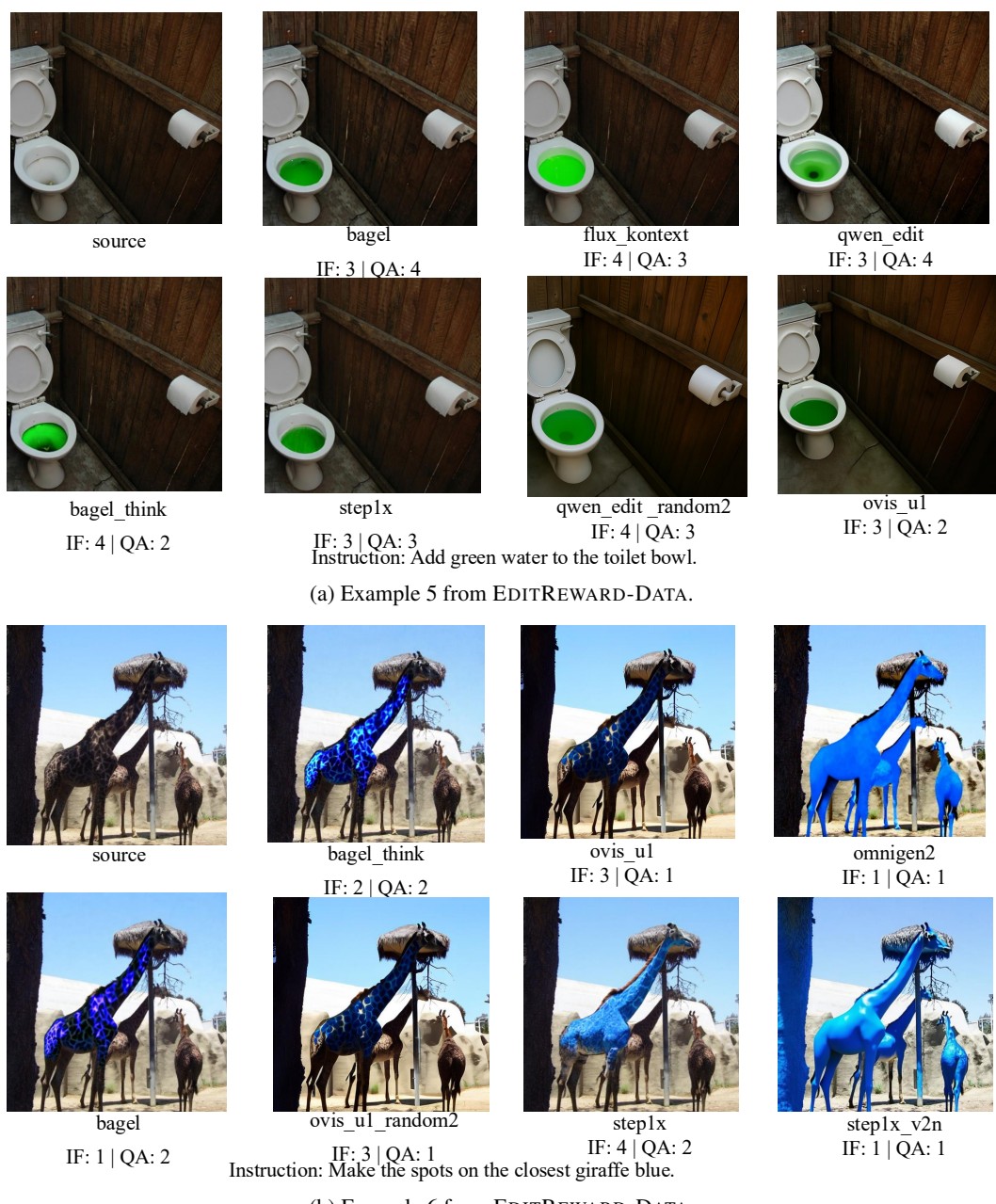

(a) Example 5 from EDITREWARD-DATA.

(b) Example 6 from EDITREWARD-DATA.

Figure 8: Representative examples from EDITREWARD-DATA, complementing Fig. 2.

## A.12 QUALITATIVE COMPARISON: BEFORE VS. AFTER EDITREWARD FILTERING

To further illustrate the qualitative improvements enabled by EditReward-based data curation, we present side-by-side comparisons of image editing results produced by the same Step1X-Edit architecture trained on two datasets: (i) the original unfiltered dataset, and (ii) the EditReward-filtered high-quality subset.

For each example, we show the **source image**, the output from the model trained on *unfiltered data* (**Before Filter**), and the output from the model trained on *EditReward-curated data* (**After Filter**). These examples cover a range of editing types, including background replacement, object insertion/removal, style and material changes, and human-centric edits.

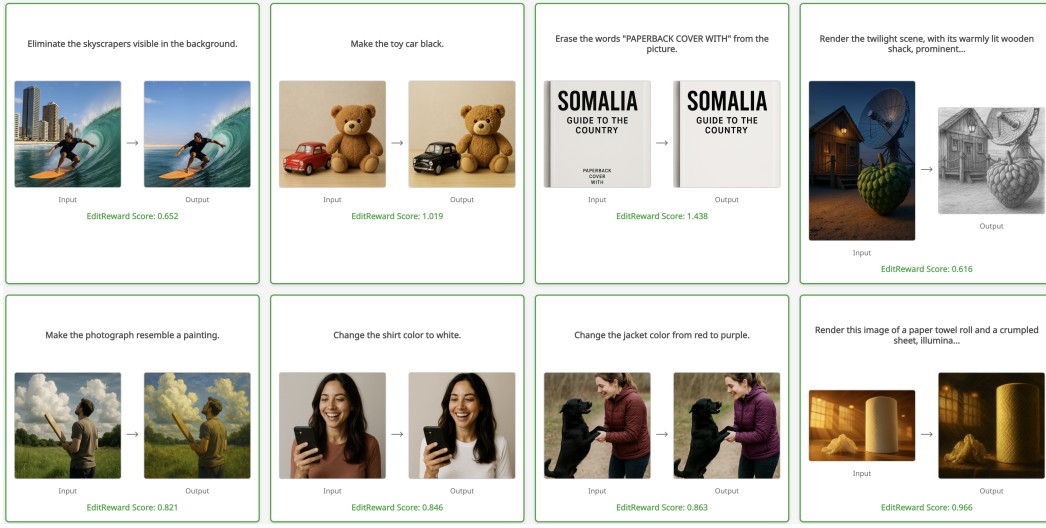

(a) Example List 1 of high-quality image editing samples.

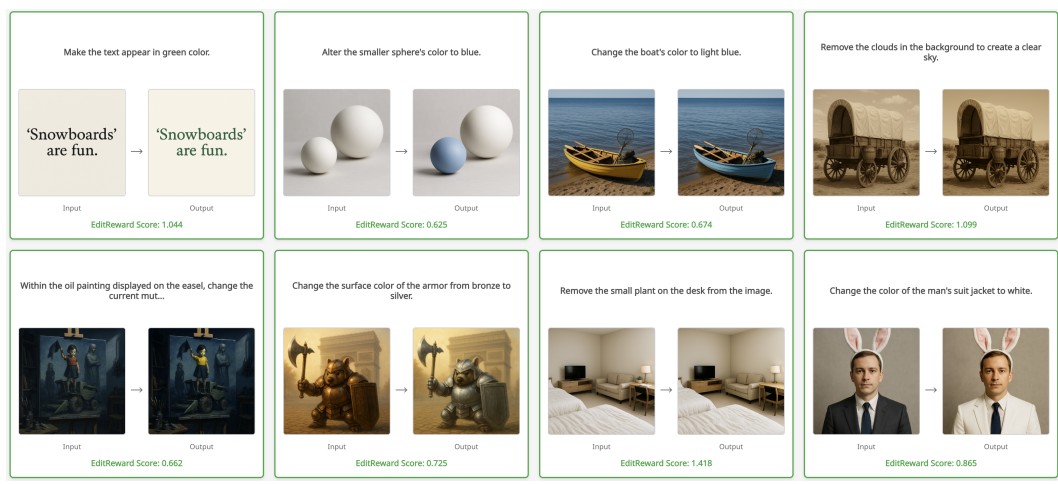

(b) Example List 2 of high-quality image editing samples.

Figure 9: Examples of high-quality image editing samples that were retained after filtering based on EditReward scores.

Qualitatively, the "After Filter" results exhibit more accurate instruction following, cleaner local modifications, fewer visual artifacts, and more coherent global integration. In contrast, models trained on unfiltered data tend to produce incomplete edits, spatial inconsistencies, or hallucinated structures. These comparisons highlight the alignment benefits of EditReward-guided data selection and validate its effectiveness in improving generation quality.

### A.13 FAILURE MODE ANALYSIS OF EDITREWARD

The following analysis moves beyond generic VLM weaknesses and focuses on specific, action-able biases exhibited by EDITREWARD when its judgment significantly deviates from the human consensus. We found two persistent failure modes:

1. **Visual Quality Bias (Color/Brightness):** We observed that EDITREWARD occasionally exhibits a **perceptual bias toward vividness**, likely inherited from its VLM pre-training. The model tends to conflate high overall visual quality with **excessive brightness or high color saturation**, resulting in inflated scores for edits that human judges deem over-processed or visually jarring.

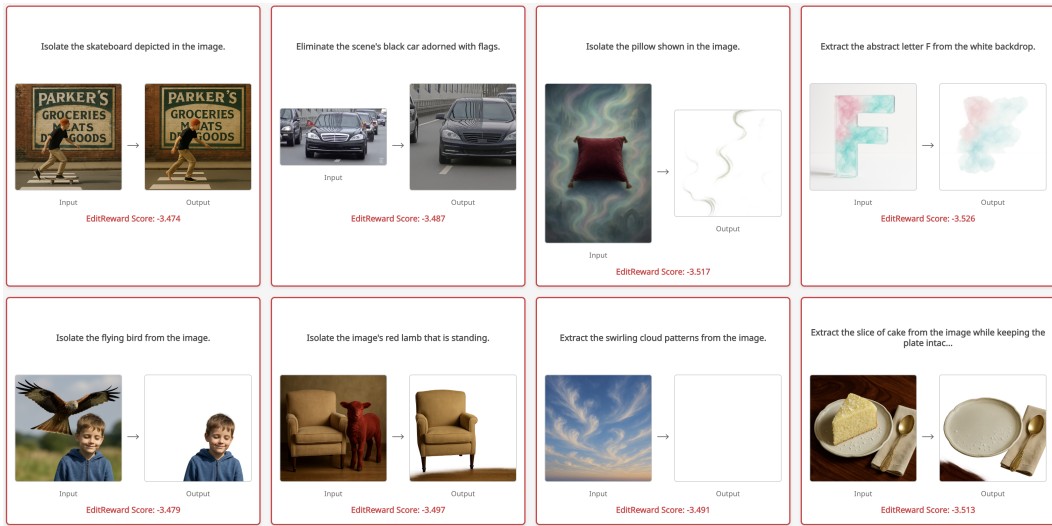

(a) Example List 1 of low-quality image editing samples.

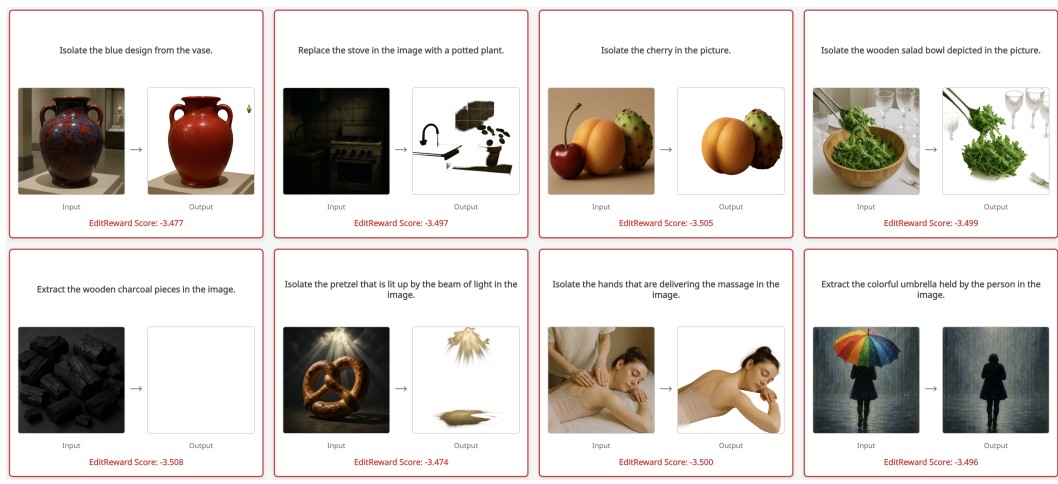

(b) Example List 2 of low-quality image editing samples.

Figure 10: Examples of low-quality image editing samples that were filtered out based on EditReward scores.

2. **Global Consistency Failure (Background Over-Editing):** Despite our explicit training on the "Exclusivity" criterion, the model sometimes gives high scores to edits where the **background or unedited regions were substantially and unnecessarily altered**. This suggests a specific weakness in balancing local edit success against the preservation mandate.

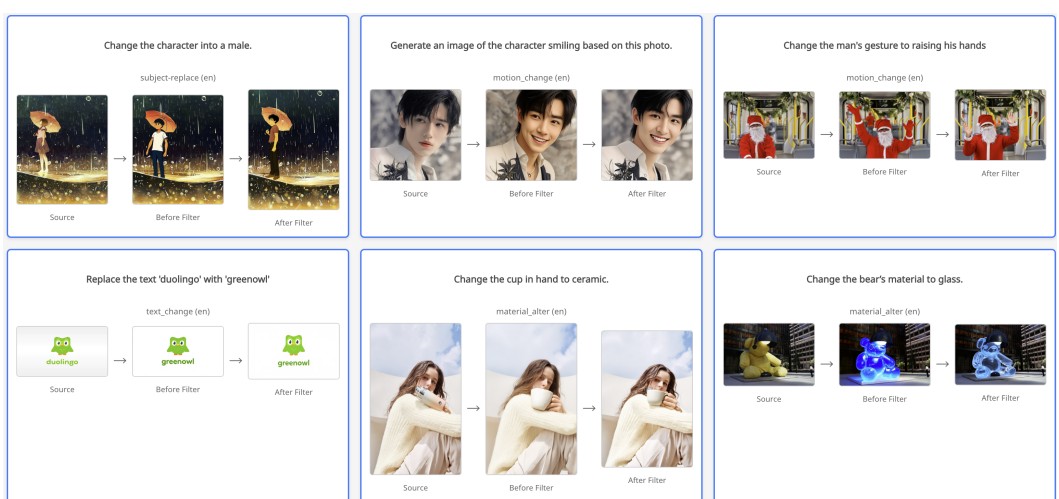

(a) Example 1: Comparison of edits produced before and after EditReward filtering.

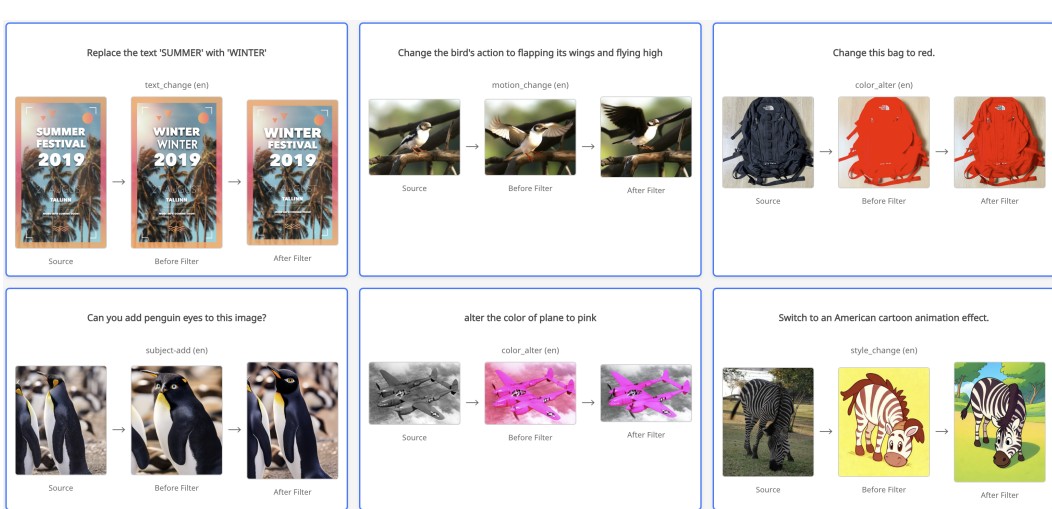

(b) Example 2: Another comparison of Step1X-Edit results trained with and without EditReward filtering.

Figure 11: Qualitative comparison of Step1X-Edit models trained on unfiltered data (Before Filter) and EditReward-curated data (After Filter). Each example shows the source image, the output before filtering, and the output after filtering. Results demonstrate that training on EditReward-filtered data produces more accurate, stable, and faithful edits.

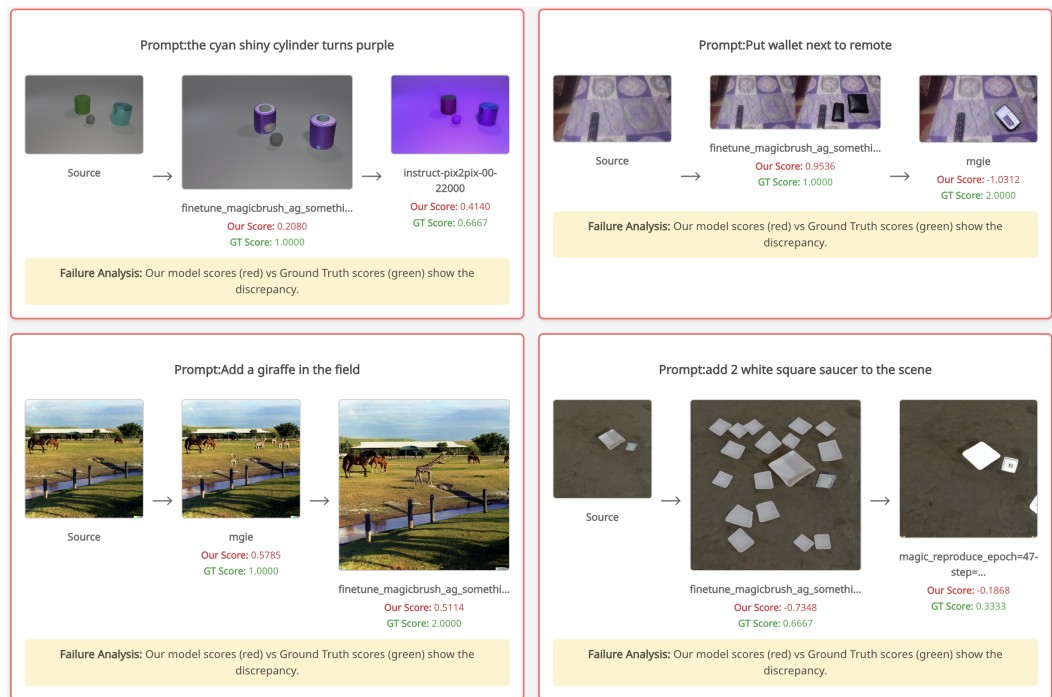

(a) Failure Mode 1: Brightness Bias. This image received a high score from EDITREWARD (e.g., Score 3.8/4.0) due to its vivid colors, even though human experts rated it lower (e.g., Score 2.0) for being over-saturated and visually implausible. This illustrates the model's tendency to reward excessive brightness.

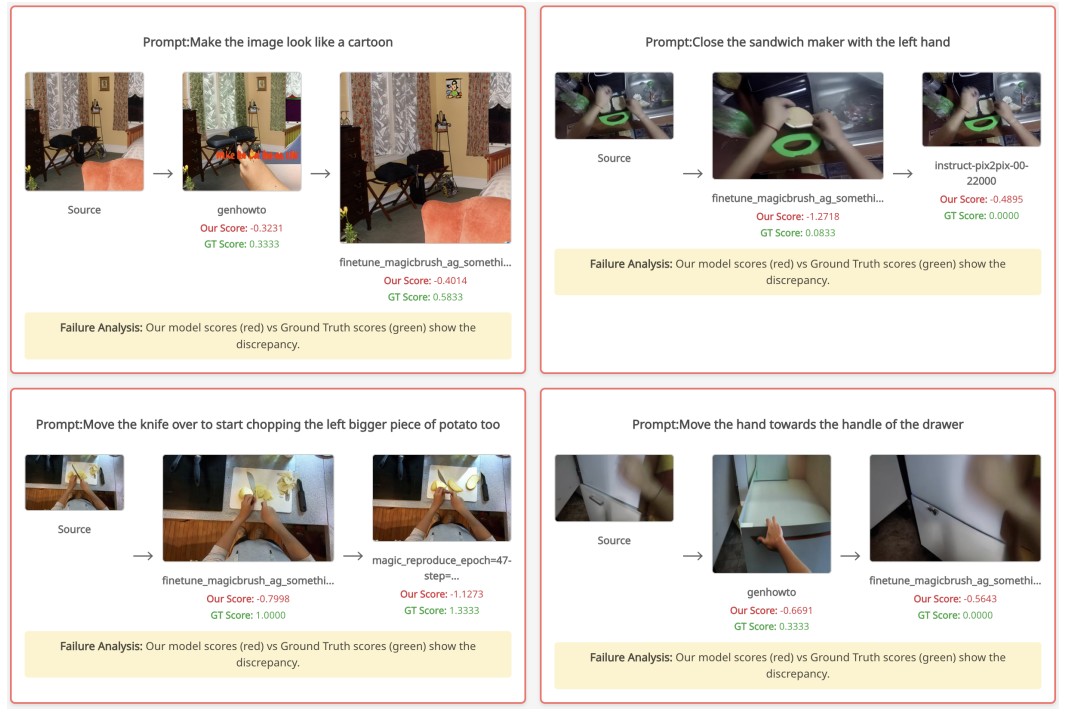

(b) Failure Mode Examplex 2: Exclusivity/Background Failure. The edit (e.g., "Change the object") was successful locally, yet the model gave it a high reward (e.g., Score 3.5/4.0) despite the background being severely altered and distorted—a clear violation of the "no unprompted changes" rule.

Figure 12: Qualitative Taxonomy of EDITREWARD's Reward Biases. These examples illustrate specific cases where EDITREWARD's high scores deviate from human consensus, demonstrating the model's inherited bias toward high color vividness and its difficulty in penalizing subtle global inconsistencies.

