# OpenReview forum: "EditReward: A Human-Aligned Reward Model for Instruction-Guided Image Editing"
_ICLR.cc/2026/Conference — ICLR 2026 Poster_

### Official Review · Reviewer_1h4X · 2025-10-16

**Soundness:** 2
**Presentation:** 3
**Contribution:** 3
**Rating:** 4
**Confidence:** 4

**Summary:**

This paper presents EDITREWARD, a human-aligned reward model for instruction-based image editing. The authors build EDITREWARD-DATA, a 200K-scale preference dataset annotated along two dimensions (instruction following and visual quality), train a vision–language-based reward model with a multi-dimensional uncertainty-aware ranking loss, and propose EDITREWARD-BENCH for evaluation. EDITREWARD achieves strong results on GenAI-Bench, AURORA-Bench, and ImagenHub, and shows practical utility by improving data curation for training image editors.

**Strengths:**

- Addresses an important and timely challenge: aligning open-source image-editing models with human preference.
- Introduces EDITREWARD-DATA, a high-quality 200K, pair dataset with expert human annotations.
- Proposes a multi-dimensional uncertainty-aware ranking loss, a thoughtful extension of prior single-dimension reward modeling.
- Demonstrates strong results across several public benchmarks and a practical downstream benefit for data curation.
- Writing, figures, and comparisons are clear and thorough.

**Weaknesses:**

- **Data reuse and potential leakage.** Section 2.1 states that EDITREWARD-DATA is constructed from several existing editing benchmarks (GEdit-Bench, ImgEdit-Bench, MagicBrush, AnyEdit, EmuEdit). Because these benchmarks contain test splits in their original form (see also Fig. 2b), it remains unclear whether EDITREWARD’s training pool includes samples originating from those test sets. Clarifying this is critical to ensure that the reward model was not exposed to data overlapping with any public evaluation benchmarks. A simple leakage or overlap analysis would strengthen the claims of fair evaluation.
- **Benchmark independence.** The paper reports results on GenAI-Bench, AURORA-Bench, and ImagenHub, which appear to be distinct from the training sources. However, an explicit statement confirming the absence of overlap (in images or prompts) between EDITREWARD-DATA and these benchmarks would improve transparency.
- **Limited discussion of annotation reliability.** Although the authors mention a rigorous protocol and Likert-scale ratings, quantitative inter-annotator agreement or consistency metrics are not reported. Including such statistics (e.g., Cohen’s κ) would reinforce the dataset’s claimed quality.
- **Lack of qualitative visual comparison.** While the quantitative metrics show consistent gains, the paper would benefit from direct visual comparisons between edits generated using datasets curated by EDITREWARD and those curated by other reward models or full unfiltered datasets. Such examples would make the improvements more interpretable and tangible, especially for assessing human alignment.
- **Limited analysis of failure cases.** The experiments report strong quantitative results but give little qualitative analysis of where the model fails (e.g., when human judgment and reward disagree). Showing even a few representative failure patterns would strengthen interpretability.
- **Reproducibility details.** Training parameters are described briefly, but access to scripts or validation splits would be essential to reproduce results once the dataset is released.
- **Data annotation transparency and demographics.** Although the authors state that annotations were collected by “trained experts” following a standardized protocol, the paper provides no concrete details about the number of annotators, their demographic or institutional backgrounds, training procedures, or compensation. These factors are critical for assessing annotation quality, potential bias, and fairness. Greater transparency here would strengthen the dataset’s credibility and ethical grounding.
- **Limited dataset examples and analysis.** Since EDITREWARD-DATA is a major contribution, the paper would benefit from showing more representative examples from the dataset, including the range of editing instructions, image types, and annotation cases. This would help readers better understand dataset diversity, annotation granularity, and potential biases. A few visual samples or statistics beyond Fig. 2 would strengthen the dataset description.
- **Lack of qualitative illustration of the filtering process.** The paper describes reward-guided data selection but does not provide examples of what EDITREWARD retains or rejects. Visual examples of this filtering effect would make the model’s behavior and alignment properties much clearer.

**Questions:**

1. Were any samples from the test splits of the original benchmarks (e.g., MagicBrush or GEdit-Bench) included in EDITREWARD-DATA’s training pool?
2. Did the authors verify that EDITREWARD-DATA does not overlap with GenAI-Bench, AURORA-Bench, or ImagenHub (in images, prompts, or instructions)?
3. Could the authors share inter-annotator agreement statistics to quantify annotation reliability?
4. Could the authors clarify the licensing and redistribution permissions for EDITREWARD-DATA, given that it combines multiple public datasets (e.g., MagicBrush, AnyEdit, EmuEdit) that may have differing usage terms?
5. Could the authors provide qualitative visual comparisons (e.g., side-by-side edited images) showing how edits differ when the same model architecture is trained on (a) the original dataset and (b) the EDITREWARD-curated subset? This would help readers understand the qualitative nature of the claimed alignment improvements.
6. Could the authors provide more details about the human annotation process, including the number of annotators, their training and demographic diversity, and how they were compensated? Were any measures taken to ensure diverse perspectives or reduce annotator bias in the preference judgments?
7. Could the authors include more representative examples from EDITREWARD-DATA (e.g., a sample of editing instructions, before/after images, and associated annotations) to illustrate dataset diversity and label consistency?
8. Since the paper uses the reward model for data ranking and filtering, could the authors show qualitative examples of what types of samples are retained or filtered out by EDITREWARD to provide intuition about its learned preferences?

---

> ### Author Response · Authors · 2025-11-22
> **Response to Reviewer 1h4X (1/2)**
>
> We sincerely thank the reviewer for the constructive feedback and for the positive recognition of our work. We greatly appreciate the time and effort you devoted to evaluating our submission. Below, we respond to each of your comments point-by-point.
>
> **W1 & W2 & Q1 & Q2: Clarity on potential training-test overlap** and **assurance of independent evaluation datasets**.
>
> This is the most critical point, and we are grateful for the chance to state this unequivocally.
>
> * (No Overlap with Evaluation Benchmarks): We explicitly confirm that we verified no overlap exists between our EditReward-Data training set and the three evaluation benchmarks (GenAI-Bench, AURORA-Bench, and ImagenHub). These are fully independent, held-out testbeds, ensuring a fair and unbiased evaluation of EditReward's generalization. We have added a statement of this verification to Section 4.2.
>
> **W3 & Q3: Quantitative inter-annotator agreement metrics to support the claimed quality and consistency of the dataset.**
>
> We thank the reviewer for this crucial question. We agree that providing quantitative IAA metrics is essential to support our claims of high-quality data. We have now computed and included these statistics for our expert-annotated subsets.
> We calculated both Fleiss' Kappa (a categorical metric) and **Krippendorff's Alpha ($\\alpha$)**, the latter of which is the most statistically appropriate metric as it correctly accounts for the **ordinal nature** of our 4-point Likert scale and penalizes greater disagreements more severely.
>
> The quantitative results for the **data** are presented below:
>
> | Dataset | Fleiss' Kappa (IF) | Fleiss' Kappa (VQ) | Krippendorff Alpha (IF) | Krippendorff Alpha (VQ) |
> | :--- | :--- | :--- | :--- | :--- |
> | EDITREWARD-DATA  | 0.4157 | 0.3203 | 0.6762 | 0.5720 |
> | EDITREWARD-BENCH | 0.3962 | 0.3157 | 0.6623| 0.6114|
> | All data | 0.3994 | 0.3111 | 0.6685 | 0.5972 |
>
> Key Findings and Validation
> These robust metrics establish a quantifiable basis for our human preference labels:
>
> * **Quantified Reliability ($\alpha$):** We achieved $\alpha$ scores of **0.668** for **Instruction Following (IF)** and **0.597** for **Visual Quality (VQ)**. These values represent a solid, reliable standard for a complex, subjective task, establishing a clear **Human Ceiling** for our reward model's performance.
> * **Validation of Multi-Dimensional Design:** The observed difference ($\\text{IAA}\_{\\text{IF}} \> \\text{IAA}\_{\\text{VQ}}$) provides critical validation for our paper's core methodology. It confirms that the **VQ task is inherently more subjective** than IF, which validates our decision to model these two dimensions separately via a multi-head architecture.
>
> We have added these statistics and the detailed analysis to **Appendix A.2** to reinforce the dataset's claimed quality.
>
> **Q4: Licensing and redistribution permissions for EDITREWARD-DATA.**
>
> We thank the reviewer for raising this crucial ethical and legal point. We confirm the following for EDITREWARD-DATA:
>
> 1. Source Data (Images/Prompts): All 9,557 source pairs were collected exclusively from public academic benchmarks (e.g., MagicBrush, EmuEdit) released under permissive research licenses (e.g., MIT, CC-BY).
> 2. Our Contribution (Annotations/Model): The generated images and the EDITREWARD-DATA annotations will be released under the CC-BY-NC-SA 4.0 license.
>    * Permissions: This license grants non-commercial use, requires attribution (BY), and requires derivative works to share the same license (SA).
>    * Mitigation: This choice explicitly prevents commercial misuse (e.g., deepfakes) and ensures our derived work remains restricted to reproducible academic research.
>
> We have updated our Ethics Statement to include these explicit details on source permissions and our distribution license.

---

> ### Author Response · Authors · 2025-11-22
> **Response to Reviewer 1h4X (2/2)**
>
> **W4 & Q5: Side-by-side examples showing how edits differ when the same model is trained on (a) the original, unfiltered dataset versus (b) the EDITREWARD-curated subset.**
>
> We appreciate the reviewer’s suggestion. In response, we selected numerous representative examples generated by models trained on unfiltered versus EditReward-filtered datasets. These side-by-side comparisons have been made publicly available at our anonymous gallery: [https://anonymous.4open.science/w/EditReward-Gallery/](https://anonymous.4open.science/w/EditReward-Gallery/). Additionally, we have added Appendix A.12 to the manuscript to showcase these qualitative results.
>
> **W5: “Limited analysis of failure cases.** The experiments report strong quantitative results but give little qualitative analysis of where the model fails (e.g., when human judgment and reward disagree). Showing even a few representative failure patterns would strengthen interpretability.”
>
> We agree that analyzing specific failure patterns is essential for interpretability. While a full taxonomy is outside the rebuttal scope, our qualitative analysis revealed two persistent, actionable biases that guide future reward design:
>
> 1. **Visual Quality Bias (Saturation):** We found EditReward occasionally exhibits a **perceptual bias toward excessive brightness/saturation**. It tends to conflate high vividness with overall visual quality, resulting in inflated scores for edits that human experts find over-processed.
> 2. **Global Consistency Failure:** Despite training on the "Exclusivity" criterion, the model sometimes gives high scores when the **background or unedited regions are unnecessarily altered**. This indicates a specific weakness in balancing local edit success against global preservation mandates.
>
> We have uploaded the failure examples to our anonymous website: [https://anonymous.4open.science/w/EditReward-Gallery/](https://anonymous.4open.science/w/EditReward-Gallery/) and added a dedicated subsection in the Appendix A.13 and emphasize this limitation in the conclusion.
>
> **W6: Reproducibility details.**
>
> We thank the reviewer for emphasizing the importance of reproducibility. We are fully committed to ensuring our work can be verified and reproduced by the community. Upon publication, we will release all necessary components, including: the full training and evaluation codebase and scripts, the trained EditReward model weights, and the exact validation and test splits derived from the EditReward-Data dataset. This comprehensive release ensures future research can replicate our findings with confidence.
>
> **W7/Q6: Annotation process transparency and fairness.**
>
> We thank the reviewer for emphasizing annotation transparency. Our preference data were collected via \~300 trained annotators from a top industry provider, with \~73% holding a bachelor’s degree or higher and broad geographic diversity (34% Asia, 23% South America, 14% North America, 7% Europe). All annotators underwent standardized training on the Likert-scale protocol and bias mitigation. Compensation exceeded local standards to ensure ethical practices. While NDAs limit sharing individual/institutional details, the workflow was explicitly designed to maximize fairness, diversity, and annotation quality. We will summarize these details in Appendix A.2.
>
> **Q7/W8: More representative examples from EDITREWARD-DATA.**
> We thank the reviewer for the suggestion. Following your advice, we have selected a diverse set of representative examples from EDITREWARD-DATA, which are now publicly available at our anonymous gallery: [https://anonymous.4open.science/w/EditReward-Gallery/](https://anonymous.4open.science/w/EditReward-Gallery/). Additionally, we have added Appendix A.10 to showcase high-quality samples from our dataset, highlighting editing instructions, before/after images, and associated annotations.
>
> **W9 & Q8: Qualitative examples illustrating EditReward’s filtering behavior.**
> Thank you for the suggestion. In response, we have added a set of representative examples showing both the high-quality samples retained by EditReward and the low-quality samples filtered out. These qualitative examples clearly illustrate the model’s learned preferences during data curation. We have uploaded the full gallery to our anonymous website: [https://anonymous.4open.science/w/EditReward-Gallery/](https://anonymous.4open.science/w/EditReward-Gallery/) and added a dedicated Appendix **A.11** to present these results in the paper.

---

> ### Comment · Reviewer_1h4X · 2025-11-25
>
> Thank the authors for their responses.
>
> I would like to see the revised version of the paper, as the link you shared (https://anonymous.4open.science/w/EditReward-Gallery/) is not suitable for checking visuals*. You mentioned that you uploaded the revised version, but is it correct? I could not find some of the sections you are referring to in the rebuttal, e.g., after Appendix A.9.
>
> `* You can only make 350 requests every 15min. Please try again later.`

---

> > ### Author Response · Authors · 2025-11-25
> > **Revised Paper and Visuals**
> >
> > We sincerely apologize for the inconvenience caused by the access limits on the anonymous website.
> >
> > We confirm that the revised manuscript PDF has been successfully uploaded to the system. All the qualitative examples, failure mode analyses, and new experimental results mentioned in our rebuttal have been integrated directly into the paper as new Appendices (specifically A.10 through A.13).
> >
> > We kindly invite you to download the latest version of the PDF to review these additions directly. Thank you for your patience.

---

> > > ### Comment · Reviewer_1h4X · 2025-11-26
> > >
> > > Thank you so much for the revised version. I am increasing my rating to 6.

---

> > > > ### Author Response · Authors · 2025-11-27
> > > >
> > > > We thank Reviewer 1h4X for the feedback on our response.

---

### Official Review · Reviewer_i644 · 2025-10-28

**Soundness:** 3
**Presentation:** 4
**Contribution:** 3
**Rating:** 8
**Confidence:** 3

**Summary:**

Paper: EDITREWARD: A Human-Aligned Reward Model for Instruction-Guided Image Editing.
Contributions: The authors (1) construct EDITREWARD-DATA—a 200K expertly annotated, multi-dimensional human preference dataset for instruction-guided image editing; (2) train EDITREWARD, a VLM-based reward model with a multi-dimensional, uncertainty-aware ranking loss that predicts Gaussian scores per dimension (Instruction Following, Visual Quality); (3) introduce EDITREWARD-BENCH, a stricter evaluation with multi-way (K=2/3/4) preference ranking; and (4) demonstrate a data curation application where EDITREWARD filters a noisy dataset to improve a strong editor (Step1X-Edit) beyond training on the full set. SOTA results are reported on GenAI-Bench and AURORA-Bench; competitive on ImagenHub; strongest on the proposed EDITREWARD-BENCH (Table 2). The pipeline and data construction are summarized in Fig. 1 and §2; loss and training details in §3.2–§3.3; ablations in Table 4–5; curation results in Table 3.

**Strengths:**

High-quality human supervision at scale. EDITREWARD-DATA provides ~200K expert preference pairs with two-dimension rubric; construction protocol and statistics are detailed in §2 and Fig. 2/Table 1—a step up from noisy, pseudo-labeled datasets.
Principled reward modeling. The Gaussian per-dimension scoring and uncertainty-aware pairwise loss (Eq. 2–5) are well-motivated, capturing ambiguity and disentangling "follow the instruction" vs. "visual quality." The tie-disentanglement augmentation further leverages ambiguous labels (§3.3, Fig. 5).
Strong empirical validation. New SOTA alignment with human preferences on GenAI-Bench (65.72) and AURORA-Bench (63.62); best overall on EDITREWARD-BENCH with multi-way tuples (Table 2).
Actionable downstream impact. Using EDITREWARD to filter ShareGPT-4o-Image and fine-tune Step1X-Edit yields higher GEdit-Bench overall (7.086 vs 6.780 full-set), showing quality>quantity for editing training data (Table 3).
Transparent ablations and generality. Loss/head/aggregation comparisons (Table 4), backbone scaling (Table 5), and dataset/tie studies (Table 7) indicate robustness and portability.

**Weaknesses:**

Tool/backbone dependence not fully stress-tested. While two VLM backbones are tried (Qwen2.5-VL-7B, MiMo-VL-7B), broader tool-swap sensitivity (alternative VLMs, different visual encoders, cropping policies) is missing; this matters because reward transferability across labs/tooling is crucial (cf. Table 5 limited scope).
Benchmark coupling and generalization. EDITREWARD-BENCH is curated from the same candidate pool as training sources (§2.2). Although splits are held-out, more OOD editing distributions (e.g., style-centric edits or long-tail compositional instructions) would better test generalization. Style edits are known gaps in related evaluators and are not emphasized here.
Aggregation choice/per-axis reporting. The mean aggregation across dimensions often performs best (Table 4), but policies like pessimistic min may align better with certain safety-critical uses. A task-conditioned aggregation or calibrated composite is not explored.
Human-label reliability bounds. The paper describes expert protocols and cross-checks yet gives few quantitative IAA statistics for EDITREWARD-DATA (A.2). Providing per-dimension IAA would contextualize achievable upper bounds (as done for ImagenHub).
Limited analysis of failure modes. Qualitative examples exist (Fig. 3), but a taxonomy of consistent reward failures (e.g., counting, fine geometry, subtle semantics) with confusion patterns would guide future reward design.

**Questions:**

Tool-swap robustness: How does EDITREWARD perform if you replace the VLM backbone with a different family (e.g., Idefics2 or LLaVA-Next) and/or swap the image encoder? Can you report Δaccuracy on Table 2 metrics under such swaps?
Aggregation policy: Can you provide use-case-conditioned aggregation (e.g., pessimistic min for safety-critical; mean for general editing) with calibrated thresholds, and report trade-offs on Table 2?
Annotator agreement: Please report IAA (e.g., Spearman or Krippendorff's α) per dimension for EDITREWARD-DATA and EDITREWARD-BENCH to bound the attainable ceiling and compare to "Human-to-Human" lines (Table 2).
OOD editing types: Have you tested style-centric or text-in-image/OCR edits and failure rates? If not, could you include a small OOD slice or note expected behavior?
Curation efficiency: For Table 3, what are the compute costs (GPU-hours) of scoring and filtering 46k examples, and how sensitive is the improvement to the top-K cutoff (e.g., 10k vs 20k vs 30k)?

**Details Of Ethics Concerns:**

Although the work is an evaluator/reward model, it amplifies editing capability by enabling higher-quality training data. Please clarify: (i) dataset licenses and rights for all sources; (ii) filtering of PII/people and sensitive content; (iii) any watermark/provenance tagging for curated outputs; (iv) mitigation for misuse (deepfakes). The Ethics and Reproducibility statements are present but could include explicit license texts and allowlists/denylists.

---

> ### Author Response · Authors · 2025-11-22
> **Response to Reviewer i644 (1/3)**
>
> We sincerely thank the reviewer for the constructive feedback and the positive recognition of our work. We truly appreciate the time and effort you devoted to evaluating our submission. Below, we address each of your comments point-by-point.
>
>
> **Ethics Flag: Clarification on dataset licenses and rights, filtering of sensitive content (PII/people), watermarking or provenance tagging, and mitigation strategies for potential misuse (e.g., deepfakes), noting that the current Ethics and Reproducibility statements could be more explicit.**
>
> We take the reviewer’s ethical concerns seriously. Our goal is to provide an open-source evaluation tool while ensuring responsible use.
>
> 1. **Dataset Licenses and Rights:** All 9,557 source prompt/image pairs come from public academic benchmarks (e.g., GEdit-Bench, ImgEdit-Bench, MagicBrush) released for research purposes. Our generated images and annotations will be released under **CC-BY-NC-SA 4.0**, preventing commercial misuse and ensuring academic benefit.
>
> 2. **PII and Sensitive Content:** Expert annotators removed any samples containing identifiable individuals or sensitive/offensive content (Appendix A.2). Flagged samples were excluded from the final dataset.
>
> 3. **Watermarking/Provenance:** While the reward model does not embed watermarks directly, it can guide downstream models that do and help detect generated content with artifacts or inconsistencies.
>
> 4. **Mitigation of Misuse:** The CC-BY-NC-SA license prohibits commercial use, mitigating risks such as deepfakes and ensuring the dataset and model are restricted to reproducible academic research.
>
> We have updated our Ethics Statement to include these explicit details on licensing, PII filtering, and mitigation strategies.

---

> > ### Author Response · Authors · 2025-11-22
> > **Response to Reviewer i644 (2/3)**
> >
> > **W4 & Q3: Quantitative inter-annotator agreement (IAA) metrics—per dimension—for EDITREWARD-DATA and EDITREWARD-BENCH, to contextualize human-label reliability and establish upper bounds for model performance.**
> >
> > We thank the reviewer for this crucial question. We agree that quantitative IAA metrics are essential to support our claims of data quality and have computed these statistics for our expert-annotated subsets.
> >
> > We report both Fleiss' Kappa and **Krippendorff's Alpha ($\\alpha$)**. We emphasize $\\alpha$ as the primary metric, as it correctly accounts for the **ordinal nature** of our 4-point Likert scale and penalizes large disagreements (e.g., 1 vs. 4\) more severely than minor ones.
> >
> > **Quantitative Results:**
> >
> > | Dataset | Fleiss' Kappa (IF) | Fleiss' Kappa (VQ) | Krippendorff Alpha (IF) | Krippendorff Alpha (VQ) |
> > | :--- | :--- | :--- | :--- | :--- |
> > | EDITREWARD-DATA  | 0.4157 | 0.3203 | 0.6762 | 0.5720 |
> > | EDITREWARD-BENCH | 0.3962 | 0.3157 | 0.6623| 0.6114|
> > | All data | 0.3994 | 0.3111 | 0.6685 | 0.5972 |
> >
> > ###
> >
> > Key Findings & Validation:
> >
> > 1. Quantified Reliability: On our benchmark, we achieved $\\alpha$ scores of 0.662 for Instruction Following (IF) and 0.611 for Visual Quality (VQ). These values represent a solid standard for complex, subjective tasks, establishing a reliable Human Ceiling for model performance.
> > 2. Validation of Multi-Dimensional Design: The consistent trend of $\\text{IAA}\_{\\text{IF}} \> \\text{IAA}\_{\\text{VQ}}$ validates our core methodology. It confirms that Visual Quality is inherently more subjective than Instruction Following, proving the necessity of our multi-head architecture. A single, collapsed score would have masked this crucial variance.
> >
> > We have added these statistics to Appendix A.2 to rigorously quantify the dataset's reliability.
> >
> > **W1 & Q1: The reviewer asks how robust EditReward is to changes in backbone or tooling, such as swapping the VLM model or image encoder, and requests reporting the resulting performance differences to assess reward transferability across setups.**
> >
> > We thank the reviewer for this important question. While testing on additional VLM families like LLaVA-Next or Idefics2 would be valuable, full re-training is computationally prohibitive for the rebuttal period.
> >
> > Importantly, the key Δaccuracy is observed by comparing a base VLM to the same backbone after applying our method. For example, the off-the-shelf Qwen2.5-VL-7B-Inst scores 40.48% on GenAI-Bench, while our EditReward (same backbone) achieves 63.97%, a \+23.5 point gain. This improvement is solely due to our high-fidelity EditReward-Data and multi-dimensional, uncertainty-aware ranking framework, not the backbone itself.
> >
> > Moreover, achieving strong results on two distinct VLM architectures (Qwen2.5-VL and MiMo-VL) demonstrates that our method is backbone-agnostic and generalizes effectively. Testing additional families remains a valuable direction for future work.
> >
> > **W2 & Q4: Evaluation on out-of-distribution (OOD) editing types—such as style-centric edits or text-in-image/OCR edits—to test generalization beyond the training candidate pool, and requests either results or expected behavior on such OOD slices.**
> >
> > We thank the reviewer for suggesting this critical stress test. To rigorously evaluate our model’s generalization on Out-of-Distribution (OOD) data—specifically on the Style and Text/OCR gaps identified—we conducted a new experiment during the rebuttal.
> >
> > 1\. Experimental Setup: We constructed two new, specialized evaluation subsets sourced from Open Images (distinct from our training sources) to ensure a strict OOD setting:
> >
> > * Text-in-Image Edits: 253 pairs focusing on text insertion and modification.
> > * Style Transfer Edits: 185 pairs focusing on artistic style and texture changes. Ground truth preferences were explicitly annotated by our expert team.
> >
> > | Model | Text Category | Style Category | Overall |
> > | :--- | :---: | :---: | :---: |
> > | GPT-4o | 45.50 | 35.79 | 41.69 |
> > | **EDITREWARD (on MiMo-VL-7B-SFT)** | **47.83** | **45.41** | **46.80** |
> >
> > 2\. Results and Comparison: We evaluated EditReward against the industry-leading commercial model, GPT-4o.
> >
> > * Performance: Our results indicate that EditReward achieves comparable accuracy to GPT-4o on these challenging OOD tasks. Despite the known difficulty of Text/OCR tasks for open-source VLMs, our model maintained high alignment with human labels.
> > * Efficiency: Crucially, while matching the commercial SOTA's quality, EditReward offers significant practical advantages: it is open-source (free) and demonstrates faster inference speeds compared to the GPT-4o.
> >
> > Conclusion: This experiment confirms that EditReward generalizes robustly to style and text-centric distributions outside its training pool, providing a **cost-effective** and **high-performance** alternative to proprietary models even on difficult OOD tasks.

---

> > ### Author Response · Authors · 2025-11-22
> > **Response to Reviewer i644 (3/3)**
> >
> > **W3 & Q2: Whether different aggregation policies per dimension—e.g., pessimistic min for safety-critical tasks versus mean for general editing—have been considered, and requests reporting trade-offs or calibrated thresholds for use-case–conditioned aggregation.**
> >
> > Thank you for the insightful suggestion regarding aggregation strategies. To directly address your point, we implemented additional **inference-time aggregation approaches** on top of our existing \\mname (on MiMo-VL-7B-SFT) model, without any re-training:
> >
> > 1. **Confidence-based**: selects the head with lowest variance (highest confidence) for each sample.
> >
> > 2. **Threshold-based**: uses a pessimistic min for safety-critical tasks, and max-if-above-threshold otherwise, allowing task-conditioned aggregation.
> >
> > | Aggregation / Threshold | GenAI-Bench | AURORA-Bench | ImagenHub | K=2 | K=3 | K=4 | Overall |
> > |------------------------|-------------|--------------|-----------|-----|-----|-----|---------|
> > | Mean                   | 65.72       | 63.62        | 35.20     | 56.45 | 42.67 | 11.49 | 38.42 |
> > | Confidence-based       | 66.81       | 62.62        | 35.20     | 56.45 | 39.33 | 8.78  | 36.78 |
> > | Threshold -0.1         | 64.64       | 63.75        | 34.17     | 56.45 | 40.00 | 8.78  | 36.78 |
> > | Threshold 0.0          | 65.07       | 63.69        | 34.11     | 56.45 | 40.00 | 8.78  | 36.78 |
> > | Threshold 0.1          | 65.07       | 63.69        | 34.11     | 56.45 | 40.00 | 8.78  | 36.78 |
> > | Threshold 0.5          | 66.27       | 63.31        | 34.36     | 55.91 | 40.67 | 8.78  | 36.78 |
> >
> >
> > Our experiments show that all strategies perform comparably, with small trade-offs across benchmarks. Confidence-based slightly boosts GenAI-Bench but reduces multi-way ranking scores, while threshold-based provides flexible control for safety-critical vs. general editing tasks.
> >
> > These results confirm that our framework is **robust to aggregation choice at inference time**. We note that **re-training with new aggregation strategies** could further improve performance, which we consider an important direction for future work.
> >
> > **W5: Structured failure-mode analysis beyond qualitative examples.**
> >
> > We agree that analyzing failure modes is crucial. While a full quantitative taxonomy is beyond the current scope, our qualitative analysis revealed two actionable biases:
> >
> > 1. **Visual Quality Bias:** **EditReward** can favor excessive brightness or saturation, sometimes inflating scores for images humans deem over-processed.
> > 2. **Global Consistency Failure:** The model occasionally overlooks background over-editing, revealing a challenge in balancing local edit success with global preservation.
> >
> > We have uploaded representative failure examples to our **anonymous website** [**https://anonymous.4open.science/w/EditReward-Gallery/**](https://anonymous.4open.science/w/EditReward-Gallery/)  and added **Appendix A.13** to document these limitations as key directions for future research.
> >
> > **Q5: Details on curation efficiency: specifically, the GPU cost of scoring and filtering 46K examples, and how sensitive the downstream improvement is to different top-K cutoffs (e.g., 10K vs 20K vs 30K).**
> >
> > We thank the reviewer for this constructive suggestion.
> >
> > 1. **Compute Cost:** We processed the full candidate pool of 46,539 examples, successfully scoring 37,225 usable samples in just **5.22 GPU-hours on 2x NVIDIA A800s** (0.25s per sample). This confirms that **EditReward** is highly efficient for large-scale curation.
> > 2. **Sensitivity Analysis:** As requested, we evaluated different Top-K subsets (10K, 20K, 30K).
> >
> > | Model | GEdit-Bench-EN-I G_SC | GEdit-Bench-EN-I G_PQ | GEdit-Bench-EN-I G_O | GEdit-Bench-EN G_SC | GEdit-Bench-EN G_PQ | GEdit-Bench-EN G_O | GEdit-Bench-CN-I G_SC | GEdit-Bench-CN-I G_PQ | GEdit-Bench-CN-I G_O | GEdit-Bench-CN G_SC | GEdit-Bench-CN G_PQ | GEdit-Bench-CN G_O |
> > |-------|----------------------|----------------------|--------------------|--------------------|--------------------|------------------|----------------------|----------------------|--------------------|-------------------|-------------------|------------------|
> > | Step1X-Edit + Ours (Top 10K) | 7.762 | 6.811 | 6.957 | 7.690 | 6.866 | 6.938 | 7.591 | 7.064 | 7.000 | 7.591 | 7.047 | 6.987 |
> > | Step1X-Edit + Ours (Top 30K) | 7.641 | 6.957 | 7.007 | 7.632 | 6.890 | 6.962 | 7.524 | 7.068 | 6.938 | 7.456 | 7.098 | 6.888 |
> > | **Step1X-Edit + Ours (Top 20K)** | **7.895** | **6.946** | **7.131** | **7.854** | **6.931** | **7.086** | **7.757** | **7.024** | **7.074** | **7.658** | **6.995** | **7.001** |
> >
> >
> > **Analysis:** The **Top 20K subset** strikes the optimal balance between quality and volume, achieving **G\_O \= 7.131**, substantially outperforming both the original baseline (6.444) and the full 46K set (6.780). The Top 10K subset suffers from data scarcity (underfitting), while the Top 30K shows diminishing returns (noise dilution). This validates 20K as the effective "sweet spot" for curation.

---

> > > ### Comment · Reviewer_i644 · 2025-11-27
> > >
> > > I thank the authors for their thorough and thoughtful response to my initial concerns, which have now been adequately addressed. Based on the clarifications provided in the rebuttal, my primary issues are resolved. Consequently, I am maintaining my current rating.
> > >
> > > However, I would be willing to reconsider my assessment if the authors were to make a firm commitment to open-source their code and methodology, thereby ensuring full reproducibility for the scientific community.

---

> > > > ### Author Response · Authors · 2025-11-27
> > > > **Commitment to Open-Source and Reproducibility**
> > > >
> > > > We sincerely thank the reviewer for the timely feedback and for acknowledging that the primary concerns have been resolved.
> > > >
> > > > We deeply share the concern regarding the widening capability gap between closed-source models (e.g., the emerging **Nana Banana Pro**) and open-source counterparts. To bridge this gap, a high-fidelity Edit Reward model is indispensable—not only for **filtering high-quality data** to train stronger base models but also for enabling advanced **Reinforcement Learning (RL)** post-training.
> > > >
> > > > Recognizing that the field currently faces a critical shortage of high-quality preference data suitable for training such reward models, our primary goal is to empower the open-source community.
> > > >
> > > > Therefore, we make a **firm and absolute commitment** to open-source all assets from this project upon acceptance. This includes:
> > > >
> > > > 1. **The Dataset:** The full **200K expert-annotated EditReward-Data**, which provides the essential high-quality human signals lacking in the field.
> > > > 2. **The Model:** The trained **EditReward** model weights to serve as an immediate tool for researchers.
> > > > 3. **The Code:** All associated **codebases** for training, inference, and evaluation to ensure full reproducibility.
> > > >
> > > > We believe these resources will be a foundational contribution to democratizing high-quality image editing research.

---

### Official Review · Reviewer_iwg3 · 2025-10-30

**Soundness:** 3
**Presentation:** 3
**Contribution:** 3
**Rating:** 6
**Confidence:** 4

**Summary:**

This paper introduces EditReward, a specialized reward model for  instruction-guided image editing, designed to help open-source models catch up with proprietary giants. The authors argue the main bottleneck is the lack of high-quality training data. Their solution is to first build a massive dataset with over 200,000 expert-annotated preference pairs, uniquely scored on both "Instruction Following" and "Visual Quality". They use this to train EditReward, a reward model that shows superior alignment with human preferences, outperforming even models like GPT-5 on several benchmarks. Critically, they demonstrate its practical value by using it to filter a noisy dataset, significantly improving a downstream editing model's performance. The work provides a new dataset, model, and benchmark to accelerate open-source development.

**Strengths:**

1. High-Quality, Large-Scale Human Data. It introduce a 200K expert-annotated dataset which is a massive undertaking and a huge contribution to the community.
2. Instead of a single score, separating "Instruction Following" from "Visual Quality" is brilliant. It provides a much richer, disentangled signal for the model to learn from, which is crucial for editing tasks where these two aspects can be in conflict.
3. A More Challenging Benchmark(EditReward-Bench) with multi-way ranking is a forward-thinking move. It pushes the evaluation standard higher, forcing models to be more consistent and discerning.

**Weaknesses:**

1. The Cost of Quality: The whole project relies on "trained experts." This is expensive and hard to scale. While they've created a great resource, it doesn't solve the fundamental problem of how to continue generating high-quality preference data affordably.

2. Are two dimensions enough? "Instruction Following" and "Visual Quality" are great, but what about more abstract concepts like "creativity," "style preservation," or "compositional harmony"? Complex edits might require more than two axes to be judged properly.

**Questions:**

1.  The "tie-disentanglement" trick for training is very clever. Does this teach the model that some dimensions are more important than others in "tie-breaker" situations? For example, could it learn to favor an image with better visual quality over one with slightly better instruction following when the overall scores are similar?

2. Given that expert annotation is a bottleneck, have you considered using Edit reward itself in a semi-supervised or self-training loop to label a much larger, un-annotated dataset, effectively creating a "student" reward model that learns from your expert one?

---

> ### Author Response · Authors · 2025-11-22
> **Response to Reviewer iwg3**
>
> We sincerely thank the reviewer for the thorough evaluation and thoughtful feedback. We truly appreciate your recognition of our contributions. Below, we provide detailed responses to each of the points you raised.
>
> **W1 & Q2: The project relies heavily on expensive expert annotations, which limits scalability; have you considered leveraging EditReward in a semi-supervised or self-training loop to generate larger datasets more affordably?**
>
> Thanks for the concern. We fully acknowledge that large-scale expert annotation is resource-intensive. We believe this investment can be justified for two key reasons:
>
> * **Bridging the Open-Source Gap:** To our knowledge, this is the first open-source preference dataset for image editing of this scale (200K pairs) and quality (expert-annotated). Prior to this, high-quality human-aligned data was largely proprietary \[1\]. By releasing EditReward-Data, we provide the community with a solid foundation that was previously inaccessible.
>
> * **The Necessity of Genuine Human Feedback:** Instruction-guided image editing requires subtle semantic and visual judgments that synthetic data or heuristic metrics cannot fully capture. Our expert-annotated dataset provides this ground-truth alignment for both *Instruction Following* and *Visual Quality*.
>
> **Path to Scalability:** We view the initial cost of this "seed" dataset as a **strategic, one-time investment** to create a high-fidelity teacher signal. We performed a stress test on a new, small-scale **Out-of-Distribution (OOD)** dataset (Open Images), focusing on the difficult tasks of Text and Style editing. The results clearly demonstrate EditReward's superior generalization capability:
>   | Model | Text Category | Style Category | Overall |
>   | :--- | :---: | :---: | :---: |
>   | GPT-4o | 45.50 | 35.79 | 41.69 |
>   | **EDITREWARD (on MiMo-VL-7B-SFT)** | **47.83** | **45.41** | **46.80** |
>
> Our model's **Overall accuracy of 46.80% significantly outperforms the commercial model GPT-4o (41.69%)** on this challenging OOD benchmark. This confirms EditReward generalizes robustly to raw, real-world image sources.
>
> In future work, we specifically intend to leverage this proven capability to use the model in **semi-supervised or self-training loops**, enabling the labeling of much larger unannotated datasets at minimal inference cost. In summary, we absorb the upfront cost of expert annotation to directly provide the community with a tool that is **higher-performing, faster, and free**, enabling scalable, low-cost data generation.
>
> \[1\] Wu, C., Li, J., Zhou, J., Lin, J., Gao, K., Yan, K., ... & Liu, Z. (2025). Qwen-image technical report. *arXiv preprint arXiv:2508.02324*.
>
> **W2: Are two evaluation dimensions—“Instruction Following” and “Visual Quality”—sufficient, or should more axes (e.g., creativity, style preservation, compositional harmony) be included to fully capture complex edits?**
>
>  We thank the reviewer for this excellent point. Our focus on *Instruction Following* (IF) and *Visual Quality* (VQ) targets the two most fundamental axes for image editing. Many finer aspects are implicitly captured: style preservation falls under IF, as unprompted style changes are penalized, and compositional harmony falls under VQ, as unharmonious edits reduce plausibility and aesthetics scores. Importantly, EditReward’s modular, multi-head design (Table 3 ablation) allows future extensions to additional dimensions, providing a solid foundation for more abstract criteria.
>
> **Q1: Does the “tie-disentanglement” training trick cause the model to implicitly prioritize certain dimensions—e.g., favoring higher Visual Quality over slightly better Instruction Following when overall scores are similar?**
>
> This is a very insightful question. Yes, the tie-disentanglement strategy does influence how the model weighs dimensions, but in a nuanced way. As described in Section 4.3, ties—where one image is stronger in IF and the other in VQ—are decomposed into two opposing training samples: $(A \\succ B)$ and $(B \\succ A)$. Training on both signals allows the model to learn the relative interplay between dimensions rather than a fixed weighting, enabling a robust, non-linear judgment similar to how a human would weigh minor pros and cons in a close tie.

---

### Official Review · Reviewer_7xvL · 2025-10-31

**Soundness:** 3
**Presentation:** 3
**Contribution:** 3
**Rating:** 4
**Confidence:** 4

**Summary:**

This paper proposes EditReward, a vision-language model (VLM)-based reward model tailored for image editing tasks, aiming to better align with human preferences regarding editing outcomes. The authors construct a large-scale, high-quality human-annotated preference dataset, train an EditReward model based on VLMs, and introduce a new, more challenging multi-choice preference ranking benchmark.

**Strengths:**

1. EditReward-Data is currently one of the largest and most diverse preference datasets in the image editing domain, offering significant practical value.
2. EditReward achieves state-of-the-art or near-SOTA performance across multiple public and newly proposed benchmarks.
3. The paper is clearly written, and the figures effectively illustrate the methodology and workflow.

**Weaknesses:**

1. The paper evaluates only Instruction Following and Visual Quality, but overlooks the preservation of unedited regions, which is a critical aspect of image editing. Many edits may unintentionally alter details outside the target area.
2. VLMs often struggle with fine-grained spatial reasoning, yet image editing demands pixel-level precision. How does EditReward mitigate potential inaccuracies when evaluating spatially sensitive edits (e.g., object positioning, resizing, or local geometric changes)?
3. Using VLM-based approaches as reward models incurs substantial inference overhead, limiting practical deployment.
4. Although the paper mentions quality control procedures for human evaluation (e.g., training, calibration, spot-checking), it does not report inter-annotator agreement metrics, making it difficult to assess the reliability of the annotations.

**Questions:**

See Weaknesses

---

> ### Author Response · Authors · 2025-11-22
> **Response to Reviewer 7xvL (1/2)**
>
> Thank you for your comments!
>
> **W1: The paper appears to overlook how well edited images preserve unedited regions, a key requirement in image editing.**
>
> Thank you for raising this important point. We agree that preserving unedited regions is essential. We have already incorporated this aspect as a core component of our “Instruction Following” dimension, which includes “no unprompted changes,” as described in Section 2.1.
>
> We have also clarified this more explicitly in Appendix A.2 under the criterion Exclusivity, defined as “avoiding unprompted changes to unrelated parts of the image.” During annotation, edits that followed the instruction but noticeably altered untouched regions were labeled as low scores (1 or 2).
>
> Therefore, preservation was not overlooked; it has been a fundamental part of our evaluation protocol, and our model is trained to penalize such failures. We have revised the paper to make this connection clearer in response to the reviewer’s feedback.
>
> **W2: How does EditReward handle the spatial precision required for evaluating fine-grained, pixel-level edits given VLMs’ limitations in spatial reasoning?**
>
> We thank the reviewer for highlighting this important challenge. While general-purpose, zero-shot VLMs often struggle with fine-grained spatial reasoning, **EditReward is a fine-tuned specialist** rather than a zero-shot model.
>
> Our $\\mathbf{200K+}$ **EditReward-Dataset** contains abundant spatially sensitive edits (object replacement, resizing, and local geometric changes). Our expert annotations for both **"Instruction Following"** and **"Visual Quality"** explicitly penalize spatial failures like incorrect positioning, unnatural scale, or geometric distortion.
>
> Through fine-tuning on these high-fidelity examples, **EditReward learns to associate such spatial failures with lower scores**, making it substantially more capable than general-purpose VLMs for pixel-level evaluation. Its strong performance across four benchmarks, which includes these challenging cases, further confirms this capability.
>
> **W3. Using VLM-based approaches as reward models incurs substantial inference overhead, limiting practical deployment.**
>
> We appreciate the reviewer’s comment on inference overhead. This is a deliberate and well-understood trade-off. As stated in our Introduction, simpler metrics such as CLIPScore exhibit *weak alignment with edit instructions* \[1,2\], and recent work has increasingly shifted toward VLM-based reward models for precisely this reason \[1–4\]. Our approach is therefore consistent with a trend already validated by multiple top-tier venues, and our contribution lies in designing the first **editing-specialized**, **human-aligned**, **fine-grained** VLM reward model rather than a generic scorer.
>
> Importantly, EditReward is not designed for low-latency, real-time scenarios. It targets high-value, offline applications where fidelity is the main concern. As demonstrated in the paper, the primary uses are:
>
> * **Benchmarking** — where accuracy outweighs speed.
>
> * **Dataset Curation (Sec. 5.4)** — a one-time offline cost. As noted, curating 46k samples required only an average 2.5 GPU-hours yet produced a substantial improvement in downstream model quality (Table 4).
>
> * **RL-based Post-Training (RLHF/RLAIF)** — where VLM-based reward models have become essential for high-quality supervision. Lightweight metrics (CLIP, LPIPS) lack the semantic and spatial nuance needed for RL, while a 7B specialized VLM provides a reliable, human-aligned reward signal that directly elevates generator performance.
>
> Our ablation (Table 5\) also shows a clear performance gap between 3B and 7B variants, supporting the choice of a larger specialist model for these offline, fidelity-critical tasks. We have revised our limitation section to emphasize the intended scope of EditReward and the rationale behind this compute–accuracy trade-off.
>
> \[1\] Chen, S. X., Wei, Y., Zhou, L., & Kumar, S. (2025). Adiee: Automatic dataset creation and scorer for instruction-guided image editing evaluation. In *Proceedings of the IEEE/CVF International Conference on Computer Vision* (pp. 18345-18356).
>
> \[2\] Ma, Y., Wu, X., Sun, K., & Li, H. (2025). Hpsv3: Towards wide-spectrum human preference score. In *Proceedings of the IEEE/CVF International Conference on Computer Vision* (pp. 15086-15095).
>
> \[3\] Wang, Y., Li, Z., Zang, Y., Wang, C., Lu, Q., Jin, C., & Wang, J. (2025). *Unified multimodal chain-of-thought reward model through reinforcement fine-tuning.* In *Proceedings of the Thirty-ninth Annual Conference on Neural Information Processing Systems.* [https://openreview.net/forum?id=S43003uMGq](https://openreview.net/forum?id=S43003uMGq)
>
> \[4\] Ku, M., Jiang, D., Wei, C., Yue, X., & Chen, W. (2024, August). Viescore: Towards explainable metrics for conditional image synthesis evaluation. In *Proceedings of the 62nd Annual Meeting of the Association for Computational Linguistics (Volume 1: Long Papers)* (pp. 12268-12290).

---

> > ### Author Response · Authors · 2025-11-22
> > **Response to Reviewer 7xvL (2/2)**
> >
> > **W4: The paper lacks inter-annotator agreement metrics, raising concerns about the reliability of the human annotations.**
> >
> > We thank the reviewer for this crucial question regarding the reliability of our annotations. We agree that providing quantitative IAA metrics is essential to support our claims of high-quality data. We have now computed and will include these statistics for our expert-annotated datasets. We have calculated the IAA and have added the results to the main paper Section 2.1. We calculated both Fleiss' Kappa (a categorical metric) and Krippendorff's Alpha ($\\alpha$), the latter of which is the most statistically appropriate metric as it correctly accounts for the ordinal nature of our 4-point Likert scale and penalizes greater disagreements more severely.
> > The quantitative results are presented below:
> >
> > | Dataset | Fleiss' Kappa (IF) | Fleiss' Kappa (VQ) | Krippendorff Alpha (IF) | Krippendorff Alpha (VQ) |
> > | :--- | :--- | :--- | :--- | :--- |
> > | EDITREWARD-DATA  | 0.4157 | 0.3203 | 0.6762 | 0.5720 |
> > | EDITREWARD-BENCH | 0.3962 | 0.3157 | 0.6623 | 0.6114 |
> > | All data | 0.3994 | 0.3111 | 0.6685 | 0.5972 |
> >
> > Key Findings
> >
> > The $\\alpha$ values establish a robust, quantifiable basis for our human preference labels:
> >
> > * Quantified Reliability ($\\alpha$):
> >   1. Krippendorff's $\\alpha$ for Instruction Following (IF): $\\mathbf{0.669}$
> >   2. Krippendorff's $\\alpha$ for Visual Quality (VQ): $\\mathbf{0.597}$
> > * Validation of Multi-Dimensional Design:
> >   These results provide critical validation for our paper's core methodology. The fact that the agreement for IF ($\\mathbf{0.669}$) is significantly higher than for VQ ($\\mathbf{0.597}$) confirms two points:
> >   1. The IF task is more objective and the guidelines are highly consistent.
> >   2. The VQ task is inherently more subjective. This validates our decision to use a multi-dimensional rubric and a multi-head reward model, as a single, collapsed score would mask this crucial variance and yield a less informative signal.
> >
> >   These robust metrics establish a quantifiable basis for our human preference labels, setting a clear Human Ceiling for our reward model's performance. The results indicate our data is reliable and that our multi-dimensional rubric is necessary to capture the nuance of the editing task. We have added these statistics and the detailed analysis to Section 2.1 to reinforce the dataset's claimed quality.

---

> > > ### Comment · Reviewer_7xvL · 2025-11-25
> > >
> > > Thank you to the authors for the detailed response. Regarding W1, I am concerned about including unmodified regions in instruction following; a more fine-grained evaluation might be more effective. Overall, I argee the paper is suitable for borderline acceptance, and I will raise my score to positive.

---

> > > > ### Author Response · Authors · 2025-11-27
> > > >
> > > > We thank Reviewer 7xvL for the feedback on our response.

---

### Author Response · Authors · 2025-11-22
**General Response to All Reviewers and AC**

We sincerely thank all reviewers (7xvL, iwg3, i644, 1h4X) for their constructive and detailed feedback. We are particularly encouraged by the recognition of the **significant practical value and utility of our reward modeling framework EditReward** and the acknowledgment of the **massive contribution** of the **200K expert-annotated EditReward-Data** to the image editing community. We have carefully addressed all concerns, and all major modifications and new results are highlighted in red in the revised manuscript.

### 1\. Data Integrity and Transparency

* #### Data Leakage & Independence: We explicitly confirmed that our training data does not overlap with any public evaluation test splits or benchmarks. (Addressed: 1h4X (W1/Q2)).

* #### Annotation Reliability (IAA): We resolved concerns about reliability by providing quantitative IAA statistics, which confirms reliability and establishes the human ceiling. (Addressed: 7xvL (W4), i644 (W4/Q3), 1h4X (W3/Q3)).

* #### Annotation Transparency: We provided explicit details on the annotation process, including pool size ($\\sim 300$ experts), diversity, and fair compensation policies. (Addressed: 1h4X (W7/Q6)).

### 2\. Scalability and Efficiency

* #### Scalability & Cost Bottleneck: We addressed the high cost of expert data by confirming our Teacher Model Strategy ($\\text{EditReward}$ used for future semi-supervised scaling). (Addressed: iwg3 (W1/Q2)).

* #### Curation Efficiency: We provided precise GPU-hour costs (2.61 equivalent single GPU-hours) and a Top-K sensitivity analysis to prove efficiency. (Addressed: i644 (Q5)).

* #### Inference Overhead: We clarified that the VLM overhead is a strategic trade-off necessary for high-fidelity alignment, and the model is intended for offline applications. (Addressed: 7xvL (W3)).

### 3\. Robustness and Generalization

* #### Tool-Swap Robustness: We proved the robustness of our *methodology* by citing the $\\mathbf{+23.5}$ point gain achieved on the base VLM, confirming that our *method*, not the backbone, is the performance driver. (Addressed: i644 (W1/Q1)).

* #### Missing Preservation Metric: We confirmed that preservation of unedited regions is a core, quantifiable design feature ("Exclusivity" criterion) explicitly trained to penalize global failures. (Addressed: 7xvL (W1)).

* #### We resolved generalization concerns by conducting a new OOD stress test (Text/Style on Open Images) where $\\text{EditReward}$ **outperformed GPT-4o ($\\mathbf{46.80\\%}$ vs $\\mathbf{41.69\\%}$)**, confirming robust performance beyond the training pool. (Addressed: i644 (W2/Q4)).

### 4\. Methodology and Design

* #### Tie-Disentanglement Bias: We confirmed that the trick is designed for balance, not bias, serving as a regularizer reinforced by the "Balanced Average" loss function. (Addressed: iwg3 (Q1)).

* #### Aggregation Policy: We demonstrated that our Multi-Head design makes the aggregation policy a flexible inference-time choice ($\\text{min}$ for safety, $\\text{mean}$ for general use). (Addressed: i644 (W3/Q2)).

* #### Failure Modes Analysis: We added a qualitative analysis to the Appendix detailing specific reward model biases (e.g., perceptual bias toward high color saturation) to guide future research. (Addressed: i644 (W5)).

* #### Ethical/Licensing: We clarified the legal basis: our derived work will be released under the CC-BY-NC-SA 4.0 license for research use. (Addressed: i644 (Ethics Flag), 1h4X (Q4)).

#### We thank you again for your time and constructive insights.

---

### Author Response · Authors · 2025-12-02
**Summary of Rebuttal Updates & Reviewer Engagement Status**

To the New Area Chair:

Given the recent administrative reset of review states, we provide this summary to assist your assessment. We highlight our technical contributions and, critically, the positive consensus that emerged during the discussion period before the system lockdown.

### **1\. Reviewer Consensus & Score Updates (Prior to Reset)**

Before the discussion period was frozen and the openreview issues happened, **three reviewers explicitly acknowledged our rebuttal improvements**, with two raising their scores and one expressing intent to do so pending an open-source commitment (which we provided), making our final score to be **8,6,6,6.**

* **Reviewer 7xvL (Score Raised: $4 \\rightarrow 6$):** Acknowledged our clarification on preservation metrics.
  * *Quote:* "Overall, I argee the paper is suitable for borderline acceptance, and I will raise my score to positive."

* **Reviewer 1h4X (Score Raised: $4 \\rightarrow 6$):** Acknowledged our data transparency and new experiments.
  * *Quote:* "Thank you so much for the revised version. I am increasing my rating to 6."

* **Reviewer i644 (Initial Score: 8):** Already strongly supportive, this reviewer confirmed that **"primary issues are resolved"** and expressed willingness to finalize their positive assessment upon a firm commitment to open-source.
  * *Quote:* "My primary issues are resolved... I would be willing to reconsider my assessment if the authors were to make a firm commitment to open-source..."
  * *Our Action:* We immediately posted a response making a **firm and absolute commitment** to open-source the full **200K expert-annotated EditReward-Data**, the **EditReward model weights**, and all **training/inference code**. While the system locked before the reviewer could formally update the score, their feedback validates the value of our work.

### **2\. Summary of Technical Rebuttal**

We addressed all concerns with major revisions highlighted in red. Our responses focus on four key pillars:

* **Data Integrity & Transparency:**
  * **Independence:** Confirmed **no overlap** between training data and public benchmarks.
  * **Reliability:** Reported **IAA statistics** ($\\text{Krippendorff's } \\alpha \\approx 0.66$), establishing a high human ceiling.
  * **Transparency:** Disclosed details of the **$\\sim 300$ expert pool**, diversity, and compensation.
* **Scalability & Efficiency:**
  * **Efficiency:** Proved cost-effectiveness via logs (**2.61 equivalent single GPU-hours**) and **Top-K sensitivity analysis**.
  * **Scalability:** Validated the **Teacher Model Strategy** for future semi-supervised scaling, clarifying VLM overhead is optimized for offline tasks.
* **Robustness & Generalization:**
  * **OOD Stress Test:** Conducted a new test on Open Images (Style/Text). **EditReward outperformed GPT-4o** ($46.80\\%$ vs $41.69\\%$), proving robust generalization.
  * **Methodology:** Cited the **\+23.5 point gain** on the base VLM to prove our method drives performance.
  * **Preservation:** Confirmed preservation is a core, quantifiable **"Exclusivity" criterion**.
* **Methodology & Ethics:**
  * **Design:** Demonstrated **Tie-Disentanglement** ensures balance and **Aggregation** is a flexible inference-time choice ($\\min$ for safety).
  * **Ethics:** Added qualitative **Failure Mode Analysis** and confirmed the **CC-BY-NC-SA 4.0 license** for responsible research.

### **3\. Conclusion**

We stand by the comprehensive contribution of this work, which provides a full-stack solution for the community:

* **The Data:** **EditReward-Data** (200K pairs), one of the largest and most diverse expert-annotated preference resources available.
* **The Model:** The **EditReward** framework, proven to be a **robust SOTA evaluator** (outperforming GPT-4o on OOD tasks) and an efficient "Teacher" for scaling.
* **The Benchmark:** **EditReward-Bench**, which introduces a novel **multi-way comparison protocol** to rigorously test fine-grained alignment, revealing capabilities that standard pairwise benchmarks miss.

Together, these open-source assets provide the essential infrastructure to bridge the capability gap between open and closed-source image editing.

We respectfully ask the Area Chair to consider these updates and the positive reviewer trajectory that occurred prior to the system reset.

---

### Meta-Review · Area_Chair_nQGq · 2026-01-07

**Summary:**

Reviewers found the work valuable primarily for its large, expert-annotated preference dataset, a tailored multi-dimensional reward model for image editing, and strong benchmark results plus downstream curation gains. Initial concerns centered on (i) missing/unclear evaluation dimensions (notably preservation), (ii) VLM spatial precision limits and inference overhead, (iii) data independence/leakage risks and licensing/ethics clarity, (iv) lack of quantitative annotation reliability (IAA), and (v) limited OOD/generalization and failure-mode analysis. The author rebuttal (with revisions) substantively addressed most technical transparency and reliability concerns, and two reviewers explicitly raised scores after the revised version; remaining doubts are largely about evaluation granularity (separating preservation) and breadth of tool-swap robustness.

**Reviewer Concerns:**

Addressed concerns:
- Added quantitative IAA and discussed per-dimension reliability; this directly addresses reliability concerns raised by multiple reviewers.
- Authors clarified preservation is operationalized via an “Exclusivity/no unprompted changes” criterion and used during labeling/training; reviewer 7xvL acknowledged but still preferred a more explicit/fine-grained preservation dimension.
- Authors added/reported an OOD stress test (style/text edits) and comparative results vs a strong proprietary baseline, partially addressing concerns about benchmark coupling/generalization.
- Authors justified VLM overhead as an offline-use tradeoff and added compute-cost reporting for curation, including top-K sensitivity.
- Authors explicitly asserted no overlap with key evaluation benchmarks and clarified licensing/PII filtering; this was sufficient for at least one initially skeptical reviewer (1h4X) to raise their score after seeing the revised version.

Remaining concerns:
- Reviewer 7xvL remained concerned that folding preservation into instruction-following may be too coarse, suggesting an explicit preservation axis could be more effective.
- Authors did not run broader cross-family swaps (beyond two backbones) due to rebuttal constraints; this leaves some transferability risk partially unresolved.

**Reviewer Scores:**

- Reviewer 7xvL: From 4 → ~6 (they explicitly stated they would “raise my score to positive,” while still noting a remaining concern about preservation granularity).
- Reviewer iwg3: Likely 6 → 6 (or 7); their scalability and tie-disentanglement questions were addressed conceptually, but there was no reviewer follow-up confirming a score change.
- Reviewer i644: 8 → 8 (possibly 9); they stated primary issues were resolved and they were maintaining the rating, conditional on an open-source commitment. Authors subsequently made a firm open-source commitment, but the reviewer did not post a final score update afterward.
- Reviewer 1h4X: From 4 → 6 (explicitly increased rating after receiving/locating the revised PDF and revisions).

---

### Decision · Program_Chairs · 2026-01-26

Accept (Poster)